# Gene network analysis identifies a central post-transcriptional regulator of cellular stress survival

**Matthew Tien[1], Aretha Fiebig[1], Sean Crosson[1,2]***

[1]Department of Biochemistry and Molecular Biology, University of Chicago, Chicago, United States; [2]Department of Microbiology, University of Chicago, Chicago, United States

**Abstract** Cells adapt to shifts in their environment by remodeling transcription. Measuring changes in transcription at the genome scale is now routine, but defining the functional significance of individual genes within large gene expression datasets remains a major challenge. We applied a network-based algorithm to interrogate publicly available gene expression data to predict genes that serve major functional roles in *Caulobacter crescentus* stress survival. This approach identified GsrN, a conserved small RNA that is directly activated by the general stress sigma factor, $\sigma^T$, and functions as a potent post-transcriptional regulator of survival across distinct conditions including osmotic and oxidative stress. Under hydrogen peroxide stress, GsrN protects cells by base pairing with the leader of *katG* mRNA and activating expression of KatG catalase/peroxidase protein. We conclude that GsrN convenes a post-transcriptional layer of gene expression that serves a central functional role in *Caulobacter* stress physiology.

DOI: https://doi.org/10.7554/eLife.33684.001

## Introduction

Organisms must control gene expression to maintain homeostasis. A common mode of gene regulation in bacteria involves activation of alternative sigma factors (σ), which redirect RNA polymerase to transcribe genes required for adaptation to particular environmental conditions. Alphaproteobacteria utilize an extracytoplasmic function (ECF) σ factor to initiate a gene expression program known as the general stress response (GSR) (*Figure 1A*). The GSR activates transcription of dozens of genes, which mitigates the detrimental effects of environmental stressors and influences the infection biology of alphaproteobacterial pathogens (reviewed in [*Fiebig et al., 2015*; *Francez-Charlot et al., 2015*]). The molecular mechanisms by which genes in the GSR regulon enable growth and survival across a chemically- and physically distinct spectrum of conditions are largely uncharacterized. Defining the functional role(s) of individual genes contained within complex environmental response regulons, such as the GSR, remains a major challenge in microbial genomics.

In the alphaproteobacterium *Caulobacter crescentus* (hereafter referred to as *Caulobacter*), strains lacking core regulators of the GSR have survival defects under multiple conditions including hyperosmotic and hydrogen peroxide stresses (*Alvarez-Martinez et al., 2007*; *Foreman et al., 2012*; *Herrou et al., 2010*; *Lourenço et al., 2011*). However, the majority of genes regulated at the transcriptional level by the *Caulobacter* GSR sigma factor, $\sigma^T$, have no annotated function or no clear role in stress physiology. While studies of transcription can provide understanding of stress responses, this approach may miss functionally important processes that are regulated at the post-transcriptional level, such as those controlled by small RNAs (sRNAs). Roles for sRNAs in bacterial stress response systems are well described (*Wagner and Romby, 2015*), but remain unexplored in the alphaproteobacterial GSR.

*For correspondence:
scrosson@uchicago.edu

**Competing interests:** The authors declare that no competing interests exist.

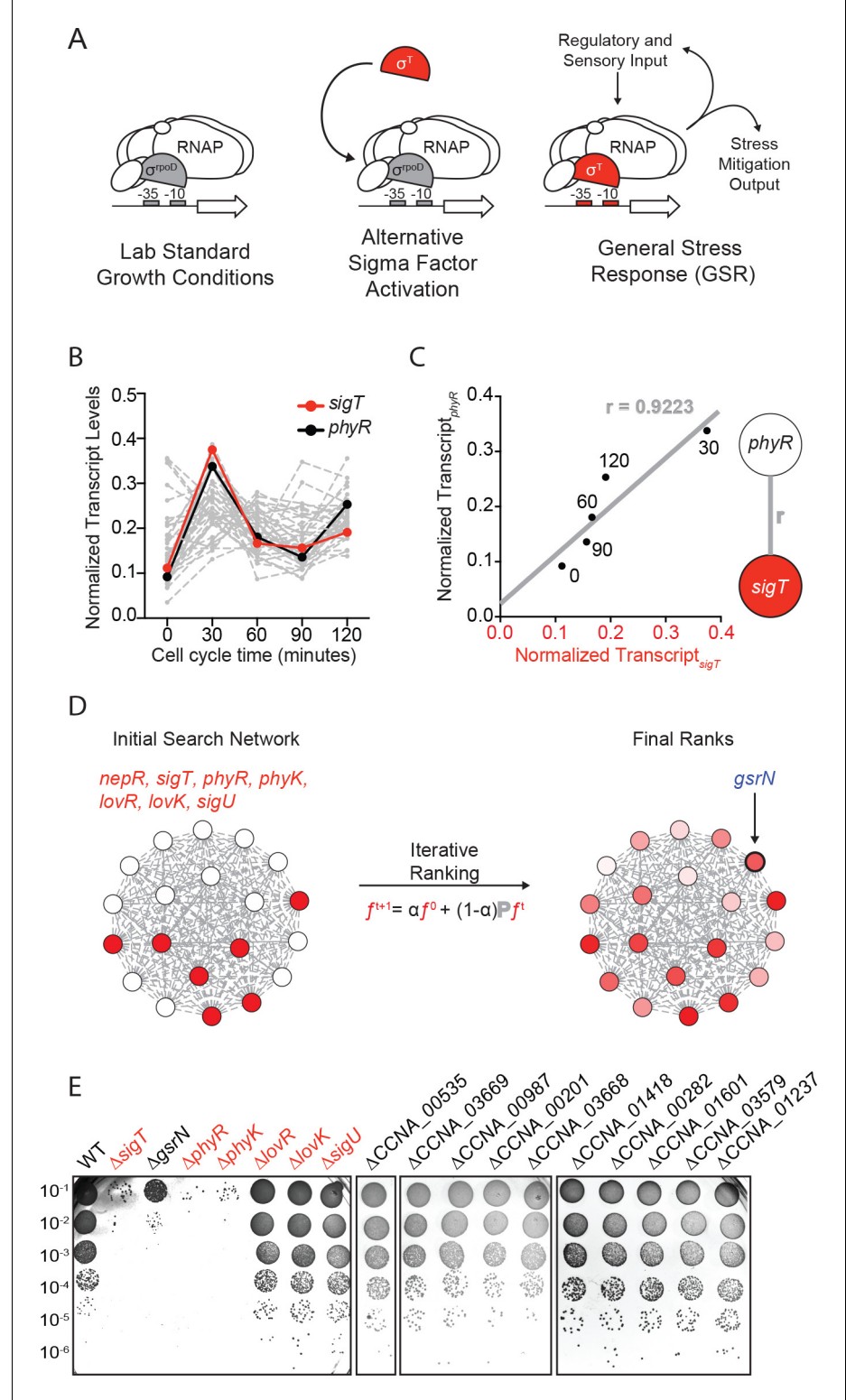

**Figure 1.** Iterative rank analysis of gene expression data identifies *gsrN*, a small RNA that confers resistance to hydrogen peroxide. (**A**) Activation of general stress response (GSR) sigma factor, $\sigma^T$, promotes transcription of genes that mitigate the effects of environmental stress and genes that regulate $\sigma^T$ activity. (**B**) Normalized transcript levels from (*Fang et al., 2013*) of known GSR regulated genes are plotted as a function of cell cycle time. The core GSR regulators, *sigT* and *phyR*, are highlighted in red and black, respectively. Data plotted from *Figure 1—source data 1*. (**C**) *sigT* and *phyR* transcript levels are correlated as a function of cell cycle progression, *Figure 1 continued on next page*

*Figure 1 continued*

Pearson's correlation coefficient $r$ = 0.92. (D) An initial correlation-weighted network was seeded with experimentally defined GSR regulatory genes (red, value = 1) (left). Final ranks were calculated using the stable solution of the iterative ranking algorithm (right). Red intensity scales with the final rank weights (*Figure 1—source data 2*). A gene encoding a small RNA, *gsrN*, was a top hit on the ranked list. (E) Colony forming units (CFU) in dilution series ($10^{-1}$ to $10^{-6}$ dilution factor) of wild-type and mutant *Caulobacter* strains after 0.2 mM hydrogen peroxide treatment for 1 hr. Red denotes core GSR regulatory genes. Black denotes known $\sigma^T$–regulated genes. GenBank locus ID is indicated for unnamed genes.

DOI: https://doi.org/10.7554/eLife.33684.002

The following source data and figure supplements are available for figure 1:

**Source data 1.** Excel file of gene expression data from (Fang et al.

DOI: https://doi.org/10.7554/eLife.33684.005

**Source data 2.** Excel file of the results from the iterative rank algorithm.

DOI: https://doi.org/10.7554/eLife.33684.006

**Figure supplement 1.** Parameter optimization of iterative rank through predicting *phyR* demonstrates edge-reduction as an important parameter (see Materials and methods - Iterative rank parameter tuning).

DOI: https://doi.org/10.7554/eLife.33684.003

**Figure supplement 2.** *gsrN* transcription is activated by $\sigma^T$ and induced in stationary phase growth.

DOI: https://doi.org/10.7554/eLife.33684.004

Regulatory roles and mechanisms of action of sRNAs are diverse: sRNAs can control gene expression by protein sequestration or by modulation of mRNA stability, transcription termination, or translation (*Wagner and Romby, 2015*). The system properties of environmental response networks are often influenced by sRNAs, which can affect the dynamics of gene expression via feedback (*Beisel and Storz, 2011*; *Mank et al., 2013*; *Nitzan et al., 2015*; *Shimoni et al., 2007*) or buffer response systems against transcriptional noise (*Arbel-Goren et al., 2013*; *Golding et al., 2005*; *Levine and Hwa, 2008*; *Mehta et al., 2008*). However, the phenotypic consequences of deleting sRNA genes are typically subtle and uncovering phenotypes often requires cultivation under particular conditions. Thus, reverse genetic approaches to define functions of uncharacterized sRNAs have proven challenging.

We applied a rank-based network analysis approach to predict functionally significant genes in the *Caulobacter* GSR regulon. The hypothesis motivating our analysis was that genes whose expression is most correlated to the core GSR regulators, calculated by iterative rank, would also be among the most important for stress response. This analysis led to the prediction that a sRNA, which we name GsrN, is a major genetic determinant of growth and survival under stress. We validated this prediction, demonstrating that *gsrN* is under direct control of $\sigma^T$ and functions as a potent post-transcriptional regulator of survival across distinct conditions including hydrogen peroxide stress and hyperosmotic shock. We developed a novel forward biochemical approach to identify direct molecular targets of GsrN and discovered that peroxide stress survival is mediated through an interaction between GsrN and the 5' leader sequence of *katG*, which activates KatG catalase/peroxidase expression. This post-transcriptional connection between $\sigma^T$ and *katG*, a major determinant of peroxide stress and stationary phase survival (*Italiani et al., 2011*; *Steinman et al., 1997*), explains the peroxide sensitivity phenotype of *Caulobacter* strains lacking a GSR system.

Finally, we demonstrate that RNA processing and sRNA-mRNA target interactions shape the pool of functional GsrN in the cell, and that changes in GsrN expression enhance expression of some proteins while inhibiting others. The broad regulatory capabilities of GsrN are reflected in the fact that a *gsrN* deletion strain has survival defects across chemically- and physically distinct stress conditions, and support a model in which the GSR initiates layered transcriptional and post-transcriptional regulatory responses to ensure environmental stress survival.

## Results

## Iterative rank analysis of gene expression data identifies a small RNA regulator of stress survival

We applied a network-based analytical approach to interrogate published transcriptomic datasets (*Fang et al., 2013*) and predict new functional genetic components of the *Caulobacter* GSR system. We organized expression data for over 4000 genes (*Figure 1B* and *Figure 1—source data 1*) to create a weighted network. In our basic network construction, each gene in the genome was represented as a node and each node was linked to every other node by a correlation coefficient that quantified the strength of co-expression across all datasets (*Figure 1C*). Within this undirected graph, we aimed to uncover a GSR clique and thus more explicitly define the core functional components of the GSR regulon.

To identify uncharacterized genes that are strongly associated with the GSR, we utilized an iterative ranking approach related to the well-known PageRank algorithm (*Brin and Page, 1998*). We defined the 'input' set as *sigT* and the experimentally defined regulators of $\sigma^T$, which include the anti-sigma factor, *nepR*, the positive two-component regulators *phyR* and *phyK,* and the negative two-component regulators *lovR* and *lovK*, as well as the paralogous sigma factor, *sigU* (in red *Figure 1D*) (*Alvarez-Martinez et al., 2007*; *Foreman et al., 2012*; *Herrou et al., 2010*; *Lourenço et al., 2011*). We then optimized parameters through a systematic self-predictability approach (*Figure 1—figure supplement 1A* and Materials and methods - Iterative rank parameter tuning) and applied iterative ranking to compute a ranked list of genes with strong associations to the input set (*Figure 1—source data 2*). We narrowed our ranked list by performing a promoter motif search on all hits to predict direct targets of $\sigma^T$. *ccna_R0081*, a gene encoding an sRNA (*Landt et al., 2008*) with a consensus $\sigma^T$ binding site (*Figure 1—figure supplement 2A*) in its promoter was a top hit in our rank list. We hereafter refer to this gene as *gsrN* (*g*eneral *s*tress *r*esponse *n*on-coding RNA) as expression of this sRNA is regulated by the GSR system (see below).

To test whether *gsrN* transcription requires the GSR sigma factor, $\sigma^T$, we generated a transcriptional reporter by fusing the *gsrN* promoter to *lacZ* ($P_{gsrN}lacZ$). Transcription from $P_{gsrN}$ required *sigT* (*Figure 1—figure supplement 2A,C*), validating *gsrN* as a bona fide member of the GSR regulon. To determine whether *gsrN* is a feedback regulator of GSR transcription, we utilized a well-characterized $P_{sigU}lacZ$ reporter (*Foreman et al., 2012*). As expected, transcription from $P_{sigU}$ required *sigT* and other GSR regulators (*phyR*, *phyK*). However, this reporter was unaffected by deletion or overexpression of *gsrN*. Furthermore, deletion or overexpression of *gsrN* did not affect activation of $P_{sigU}$ transcription upon addition of 150 mM sucrose, a known GSR inducer (*Figure 1—figure supplement 2D*). We conclude *gsrN* is activated by $\sigma^T$, but does not feedback to control GSR transcription.

We next tested whether *gsrN* plays a role in stress survival. We subjected strains lacking *gsrN* or the core GSR regulators, *sigT*, *phyR*, or *phyK*, to hydrogen peroxide, a known stress under which GSR regulatory mutants have a survival defect. $\Delta sigT$, $\Delta phyR$, and $\Delta phyK$ strains had a $\approx$4-log decrease in cell survival relative to wild type after exposure to hydrogen peroxide, as previously reported (*Alvarez-Martinez et al., 2007*; *Foreman et al., 2012*; *Lourenço et al., 2011*). Cells lacking *gsrN* ($\Delta gsrN$) had a $\approx$3-log viability defect relative to wild type (*Figure 1E* and *Figure 1—figure supplement 2B*). Insertion of *gsrN* with its native promoter at the ectopic *vanA* locus fully complemented the peroxide survival defect of $\Delta gsrN$ (*Figure 2A* and *Figure 2—figure supplement 1C*). These data provide evidence that *gsrN* is a major genetic contributor to cell survival upon peroxide exposure. To query if other $\sigma^T$-regulated genes are important for peroxide survival, we selected 10 additional genes that are strongly regulated by $\sigma^T$ based on past transcriptome studies (*Alvarez-Martinez et al., 2007*; *Foreman et al., 2012*) and generated strains harboring single, in-frame deletions of these genes. The functions of these 10 genes are unknown: six encode conserved hypothetical proteins; two encode predicted outer membrane proteins; one encodes a cold shock protein, and one encodes a ROS/MUCR transcription factor. None of these additional deletion strains were sensitive to hydrogen peroxide (*Figure 1E* and *Figure 1—figure supplement 2B*).

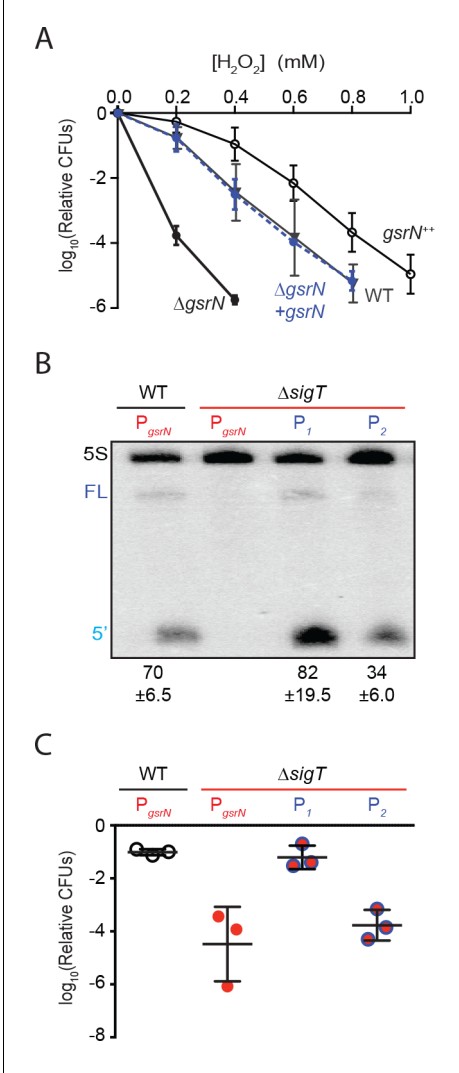

**Figure 2.** GsrN is necessary for hydrogen peroxide stress survival, and expression of GsrN is sufficient to confer peroxide protection in a *sigT* null background. (A) *Caulobacter* wild type (WT), *gsrN* deletion (Δ*gsrN*), complementation (Δ*gsrN + gsrN*), and *gsrN* overexpression (*gsrN*++) strains were subjected to increasing concentrations of hydrogen peroxide for 1 hr and titered on nutrient agar. Complementation and overexpression strains carry plasmids with one or three copies, respectively, of *gsrN* with its native promoter integrated at the ectopic *vanA* locus (see *Figure 2—figure supplement 1A* for details). Δ*gsrN* and WT strains carried the empty plasmid (pMT552) integrated at the *vanA* locus. Log$_{10}$ relative CFU (peroxide treated/untreated) is plotted as a function of peroxide concentration. Mean ±SD, n = 3 independent replicates. (B) Northern blot of total RNA isolated from WT and Δ*sigT* strains expressing *gsrN* from its native promoter (P$_{sigT}$) or from two constitutive σ$^{RpoD}$ promoters (P$_1$ or P$_2$); probed with $^{32}$P-labeled oligonucleotides specific for GsrN and 5S rRNA as a loading control. Labels on the left refer to 5S rRNA (5S

*Figure 2 continued on next page*

## Expression of GsrN confers protection from peroxide stress

Results outlined above demonstrate that *gsrN* is necessary for hydrogen peroxide stress survival. To assess the effects of *gsrN* overexpression, we inserted constructs containing either one or three copies of *gsrN* under its native promoter into the *vanA* locus of wild-type and Δ*gsrN* strains (*Figure 2—figure supplement 1A*). We measured GsrN expression directly in these strains by Northern blot (*Figure 2—figure supplement 1B*) and tested their susceptibility to hydrogen peroxide (*Figure 2A*). Treatment with increasing concentrations of hydrogen peroxide revealed that strains overexpressing *gsrN* have a survival advantage compared to wild type. Measured levels of GsrN in the cell directly correlated (*r = 0.92*) with cell survival providing evidence that *gsrN* confers dose dependent protection from peroxide stress over the measured range (*Figure 2—figure supplement 1C*).

Given that σ$^T$ regulates many genes, we sought to test if *gsrN* expression was sufficient to mediate cell survival under peroxide stress in a strain lacking σ$^T$ (and thus GSR transcription). To decouple *gsrN* transcription from σ$^T$, we constitutively expressed *gsrN* from promoters (P1 and P2) controlled by the primary sigma factor, RpoD, in a strain lacking *sigT* (*Figure 2—figure supplement 1A*). *gsrN* expression from P1 was 15% higher, and expression from P2 50% lower than *gsrN* expressed from its native σ$^T$-dependent promoter (*Figure 2B*). Expression of *gsrN* from P1, but not P2, rescued the Δ*sigT* peroxide survival defect (*Figure 2C*). We conclude that *gsrN* is a major genetic determinant of hydrogen peroxide survival regulated downstream of σ$^T$ under these conditions. Consistent with the dose dependent protection by GsrN, these data demonstrate that a threshold level of *gsrN* expression is required to protect the cell from hydrogen peroxide.

## GsrN is endonucleolytically processed into a more stable 5' isoform

A notable feature of GsrN is the presence of two isoforms by Northern blot. Probes complementary to the 5' portion of GsrN reveal full-length (≈100 nucleotide) and short (51 to 54 nucleotides) isoforms while probes complementary to the 3' portion reveal mostly full-length GsrN (*Figure 3A* and *Figure 3—figure supplement 1A*). Smaller 3' isoforms are apparent as minor species when high concentrations of total RNA

*Figure 2 continued*

in black), full-length GsrN (FL in dark blue), and the 5'isoform of GsrN. (5' in cyan) Quantified values are mean ±SD of normalized signal, n = 3 independent replicates. (C) Relative survival of strains in (B) treated with 0.2 mM hydrogen peroxide for 1 hr normalized as in (A). Mean ±SD from three independent experiments (points) is presented as bars.

DOI: https://doi.org/10.7554/eLife.33684.007

The following figure supplement is available for figure 2:

**Figure supplement 1.** GsrN-dependent cell protection under oxidative stress is dose dependent.

DOI: https://doi.org/10.7554/eLife.33684.008

are probed (*Figure 3—figure supplement 1A*). Two isoforms of GsrN are also evident in RNA-seq data (*Figure 3—figure supplement 1B*).

The short isoform of *gsrN* could arise through two biological processes: alternative transcriptional termination or endonucleolytic processing of full-length GsrN. To begin to discriminate between these two possibilities, we inhibited transcription with rifampicin, and monitored levels of both GsrN isoforms over time. Full-length GsrN decayed exponentially with a half-life of ~105 s (*Figure 3—figure supplement 1C,D*). The 5' isoform increased in abundance for several minutes after treatment, concomitant with the decay of the full-length product. This observation is consistent with a model in which the 5' isoform arises from the cleavage of the full-length product.

To identify potential endonucleolytic cleavage sites, we conducted primer extension assays to map the 5' termini of the isoforms. Primer extension binding sites are shown in (*Figure 3E*). Extension from an oligo complementary to the 5' portion of GsrN confirmed the annotated transcriptional start site (*Figure 3C*). Extension from the 3' portion identified two internal 5' ends (*Figure 3D*). The positions of these internal 5' ends are consistent with two small bands observed on Northern blots of high concentrations of total RNA hybridized with the 3' probe (*Figure 3—figure supplement 1A*). The terminus around C53 corresponds to a potential endonucleolytic cleavage site that would generate the abundant stable 5' isoform (*Figure 3B*).

To directly test if the 5' termini identified by primer extension and supported by Northern blotting reflect termini generated by cleavage or by transcription initiation, we implemented Rapid Amplification of cDNA 5'ends (5'RACE) with differential tobacco acid pyrophosphatase (TAP) treatment (*Bensing et al., 1996*). In this protocol, cDNAs with 5' termini formed by transcription initiation are amplified only with TAP treatment, whereas those with ends generated by processing are amplified with or without TAP treatment. We were able to clone cDNAs corresponding to both 3'isoforms in both TAP-treated and untreated total RNA samples of *gsrN*++ cultures (Figure 3—figure supplement 3E). The ends of these clones (T54 and T64) are consistent with the ends mapped by primer extension. Together, these results support a model in which the full-length GsrN transcript is endonucleolytically processed into a stable 5' isoform and a less stable 3' isoform.

## Hfq stabilizes full-length GsrN

sRNAs are often associated with the bacterial RNA chaperone, Hfq, and in some cases also associated with another RNA chaperone ProQ (*Gottesman and Storz, 2011*; *Smirnov et al., 2016*; *Vogel and Luisi, 2011*). *Caulobacter*, however, does not have an obvious ProQ homologue. To test the influence of *hfq* on GsrN, we created deletion (Δ*hfq*) and overexpression (*hfq*++) strains. We observed that *hfq* affects GsrN processing. In Δ*hfq* strains, full-length GsrN is undetectable by Northern blot, even when *gsrN* is overexpressed (*Figure 3—figure supplement 2*). Conversely, overexpression of *hfq*++ results in increased levels of full-length GsrN that exceed levels of the 5'isoform. We conclude that *hfq* influences the processing of GsrN in vivo. We note that the growth rate of both these strains is significantly attenuated in defined M2X medium. Moreover, large granules have been observed microscopically in Δ*hfq* strains (*Irnov et al., 2017*). Given the pleiotropic consequences of *hfq* deletion, stress survival phenotypes for these strains are difficult to interpret.

## 5' end of GsrN is necessary to mediate peroxide survival

To test the function of the 5' end of GsrN, we integrated a *gsrN* allele that contains only the first 58 nucleotides (Δ59–106), and lacks the transcriptional terminator (*gsrN*Δ3') into the *vanA* locus (*Figure 4A*). This short *gsrN* allele complemented the Δ*gsrN* peroxide survival defect (*Figure 4B*). The *gsrN*(Δ3') allele produced a 5' isoform that was comparable in size and concentration to the

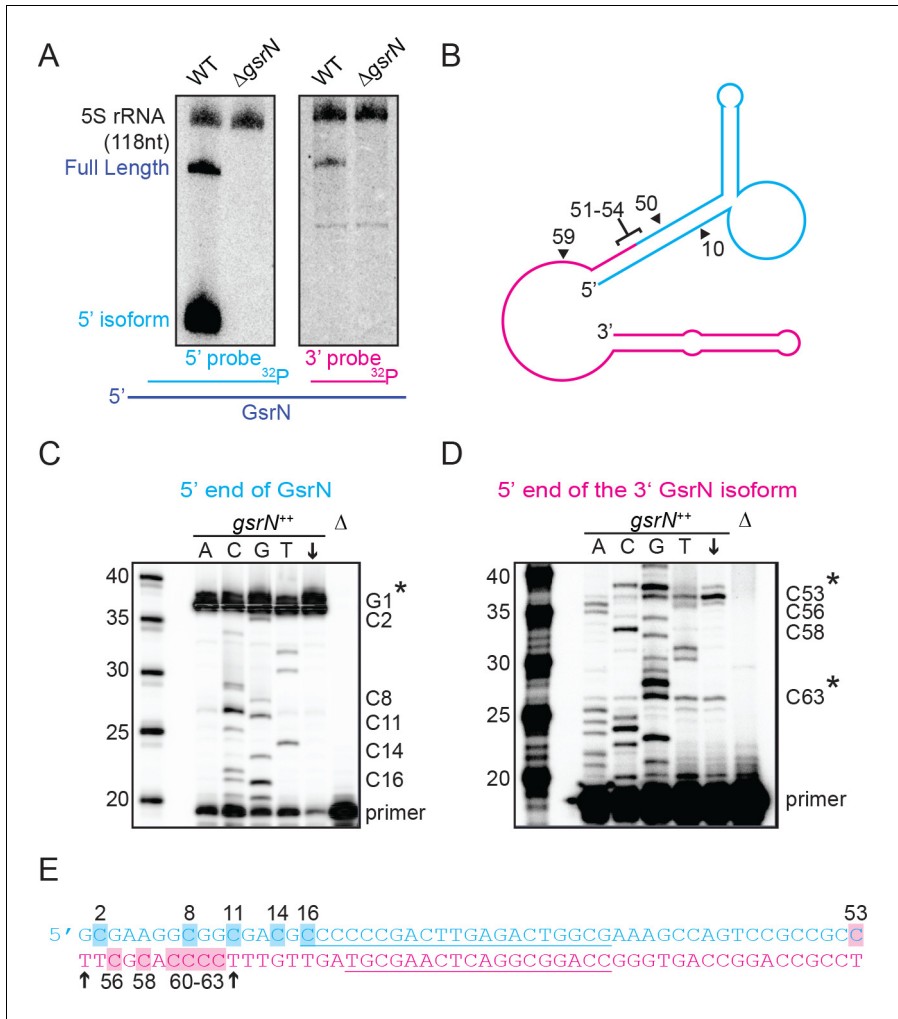

**Figure 3.** Full-length GsrN is endonucleolytically processed into a stable 5' isoform. (A) Northern blots of total RNA from wild-type and ΔgsrN cells hybridized with probes complementary to the 5'end (left) or 3' end (right) of GsrN, and to 5S rRNA as a loading control. (B) Predicted secondary structure of full-length GsrN using RNA-specific folding parameters (*Andronescu et al., 2007*). Cyan indicates the 5' end of GsrN determined by primer extension. Pink represents the 3' end. Nucleotide positions labeled with arrows provides context for the mutants in *Figure 4*. (C) Primer extension from total RNA extracted from gsrN++ and ΔgsrN (negative control) cultures (OD660 ≈ 1.0, a condition in which GsrN levels were observed to be the highest). Sequence was generated from a radiolabeled oligo anti-sense to the underlined cyan sequence in (E). Sanger sequencing control lanes A, C, G, and T mark the respective ddNTP added to that reaction to generate nucleotide specific stops. C' labels on the right of the gel indicate mapped positions from the 'G' lane. Arrow indicates lane without ddNTPs. Asterisk indicates positions of 5' termini. (D) Primer extension from RNA samples as in (C). Sequence was extended from a radiolabeled oligo anti-sense to the underlined pink sequence in (E). (E) GsrN coding sequence. Cyan and pink indicate the predicted 5' and 3' isoforms, respectively. Primers binding sites used for primer extension in (C) and (D) are underlined. Highlighted C positions correspond to ddGTP stops in the 'G' extensions. Black arrowheads correspond to the termini identified by 5'RACE (*Figure 3—figure supplement 1*).

DOI: https://doi.org/10.7554/eLife.33684.009

The following figure supplements are available for figure 3:

**Figure supplement 1.** The 5' isoform of GsrN arises from endonucleolytic processing and is the most abundant form of GsrN.

DOI: https://doi.org/10.7554/eLife.33684.010

**Figure supplement 2.** Hfq stabilizes full-length GsrN.

DOI: https://doi.org/10.7554/eLife.33684.011

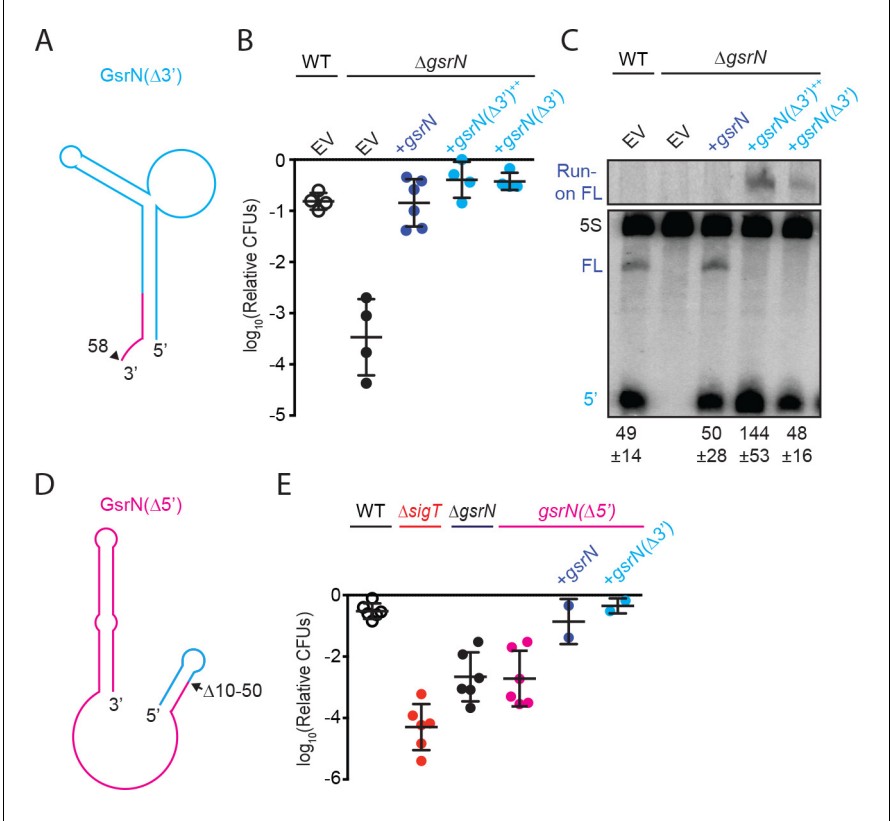

**Figure 4.** 5' end of GsrN is necessary and sufficient for peroxide survival. (A) Schematic diagram of GsrN(Δ3'), which lacks nucleotides 59–106, which includes the intrinsic terminator. Nucleotide positions are highlighted in *Figure 3B*. (B) Relative survival of strains treated with 0.2 mM hydrogen peroxide for 1 hr. WT and ΔgsrN strains carry empty intergrated plasmids (EV) or integrated plasmids harboring full-length *gsrN*, *gsrN*(Δ3'), or multiple copies of *gsrN*(Δ3') (labeled *gsrN*(Δ3')++). Bars represent mean ±SD from four independent experiments (points). (C) Northern blot of total RNA from strains in panel 3B harvested during exponential growth phase. Blots were hybridized with probes complementary to the 5' end of GsrN and 5S rRNA. Mean ±SD of total GsrN signal from three independent samples. (D) Schematic diagram of GsrN(Δ5'), which lacks nucleotides 10–50, but contains the intrinsic terminator of GsrN (the terminal 3' hairpin). Nucleotide positions are highlighted in *Figure 3B*. (E) Relative survival of strains treated with 0.2 mM hydrogen peroxide for 1 hr. Genetic backgrounds are indicated above the line; the GsrN(Δ5') strain was complemented with either *gsrN* (dark blue) or GsrN(Δ5') (cyan). Bars represent mean ±SD from at least two independent experiments (points).

DOI: https://doi.org/10.7554/eLife.33684.012

wild-type 5' *gsrN* isoform. Since the transcriptional terminator of *gsrN* was removed, we also observed a ~ 200 nt run-on transcript from *gsrN*(Δ3') (*Figure 4C*).

To test whether the 5' end of GsrN is necessary for peroxide stress survival, we deleted nucleotides 10 to 50 of *gsrN* at its native locus (*Figure 4D*). The *gsrN*(Δ5') strain had a peroxide viability defect that was equivalent to Δ*gsrN*. Ectopic expression of either full-length *gsrN* or *gsrN*(Δ3') in the *gsrN*(Δ5') strain complemented this peroxide survival defect (*Figure 4E*).

## Several RNAs, including *katG* mRNA, co-purify with GsrN

We developed a forward biochemical approach to identify molecular partners of GsrN. The <u>P</u>seudo<u>monas</u> phage<u>7</u> (PP7) genome contains hairpin (PP7hp) aptamers that bind to PP7 coat protein (PP7cp) with nanomolar affinity (*Lim and Peabody, 2002*). We inserted the PP7hp aptamer into multiple sites of *gsrN* with the goal of purifying GsrN with its interacting partners from *Caulobacter* lysates by affinity chromatography (*Figure 5A*), similar to an approach used by (*Hogg and Collins, 2007*; *Said et al., 2009*). PP7hp insertions at the 5' end of *gsrN* and at several internal nucleotide positions (37, 54, 59, 67, and 93nt) were functionally assessed (*Figure 5—figure supplement 1A*).

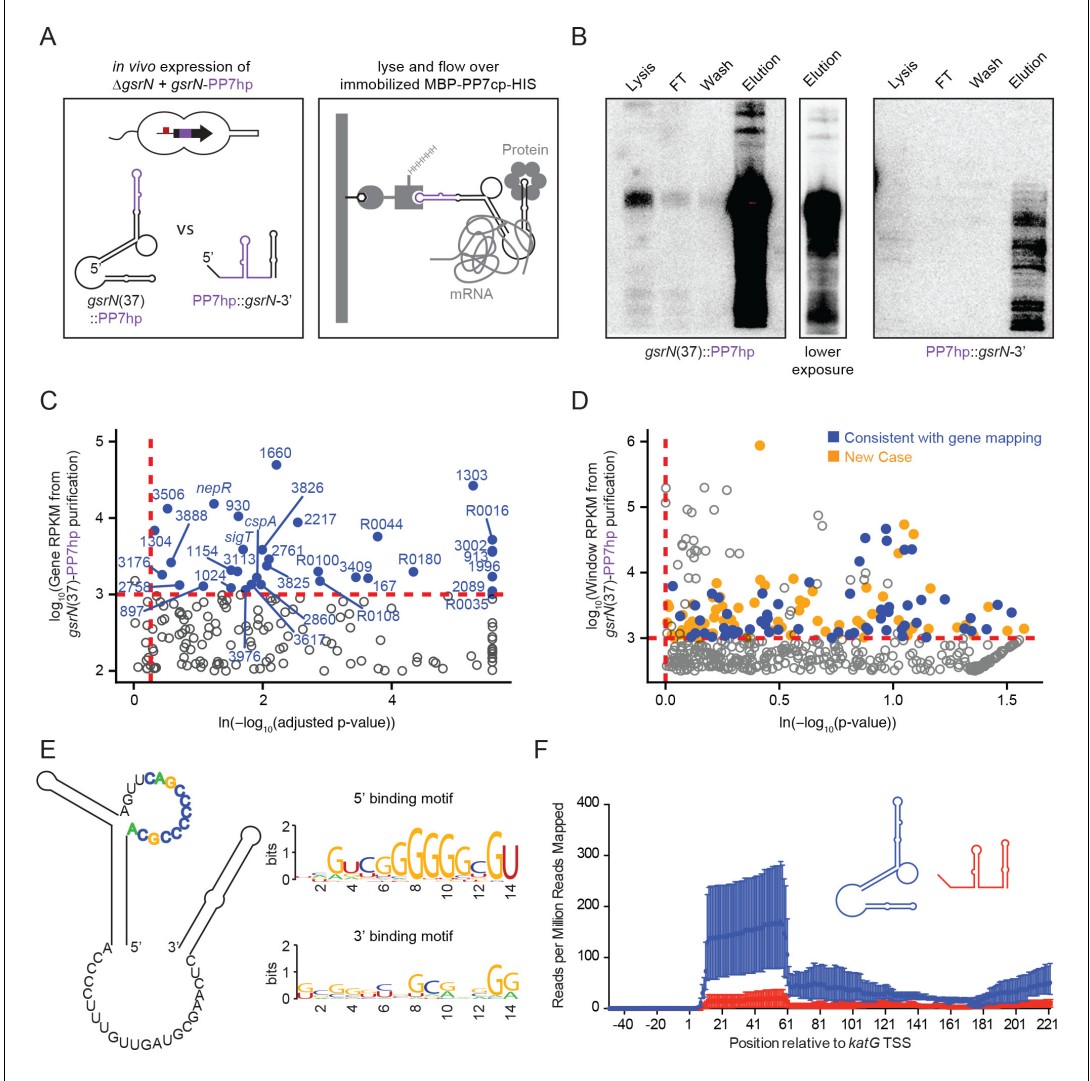

**Figure 5.** GsrN co-purifies with multiple RNAs, including catalase/peroxidase *katG* mRNA. (**A**) GsrN-target co-purification strategy. GsrN(black)-PP7hp (purple) fusions were expressed in a Δ*gsrN* background. PP7 RNA hairpin (PP7hp) inserted at nucleotide 37 (*gsrN*(37)::PP7hp) was used as the bait. PP7hp fused to the 3′ hairpin of *gsrN* (PP7hp::*gsrN*-3′)served as a negative control. Stationary phase cultures expressing these constructs were lysed and immediately flowed over an amylose resin column containing immobilized PP7hp binding protein (MBP-PP7cp-His). (**B**) GsrN-PP7hp purification from strains bearing *gsrN*(37)::PP7hp (left) and PP7hp::*gsrN*-3′ (right) was monitored by Northern Blot with probes complementary to 5′ end of GsrN and PP7hp, respectively. Lysate, flow through (FT), buffer wash, and elution fractions are blotted. Approximately 1 μg RNA was loaded per lane, except for buffer wash (insufficient amount of total RNA). (**C**) Annotation-based analysis of transcripts that co-purify with *gsrN*(37)::PP7hp (*Figure 5—source data 1*). Log$_{10}$ reads per kilobase per million reads (RPKM) is plotted against the ln(-log$_{10}$(false discovery rate corrected p-value)). Dashed red lines mark the enrichment co-purification thresholds. Genes enriched in the *gsrN*(37)::PP7hp purification compared to PP7hp::*gsrN*-3′ are blue; labels correspond to gene names or *C. crescentus* strain NA1000 CCNA GenBank locus ID. Data represent triplicate purifications of *gsrN*(37)::PP7hp and duplicate PP7hp::3′GsrN control purifications. Log adjusted p-values of zero are plotted as 10$^{-260}$. (**D**) Sliding-window analysis of transcripts that co-purify with *gsrN*(37)::PP7hp (*Figure 5—source data 2*). Points represent 25 bp genome windows. RPKM values for each window were estimated by EDGE-pro; p-values were estimated by DESeq. Windows that map to genes identified in (**C**) are blue. Orange indicates windows with significant and highly abundant differences in mapped reads between *gsrN*(37)::PP7hp fractions and the PP7hp::*gsrN*-3′ negative control fractions. Dashed red lines denote cut-off value for windows enriched in the *gsrN*(37)::PP7hp fractions. Grey points within the dashed red lines are signal that mapped to rRNA. (**E**) Predicted loops in GsrN accessible for mRNA target base pairing are emphasized in colored texts. A putative mRNA target site complementary to a cytosine-rich tract in the 5′ GsrN loop is represented as a sequence logo. Similar logo was generated for the target site sequences complementary to the 2$^{nd}$ exposed region in the 3′ end of GsrN. Logo was generated from IntaRNA 2.0.2 predicted GsrN-binding sites in transcripts enriched in the *gsrN*(37)::PP7hp pull-down. 5′ binding motif is present in 32 of the transcripts identified in (**C**) and (**D**) and 3′ binding motif is present in 27 of the transcripts identified in (**C**) and (**D**). (**F**) Density of reads mapping to *katG* that co-purified with *gsrN*(37)::PP7hp (blue) and PP7hp::*gsrN*-3′ (red). Read density in each dataset represents read coverage at each nucleotide divided by the number of million reads mapped in that data set. Data represent mean ±SD of three replicate *gsrN*(37)::PP7hp and two replicate PP7hp::*gsrN*-3′ purifications.

*Figure 5 continued on next page*

*Figure 5 continued*

DOI: https://doi.org/10.7554/eLife.33684.013

The following source data and figure supplement are available for figure 5:

**Source data 1.** Excel file of the output from Rockhopper analysis (*Tjaden, 2015*) on the RNA-Seq samples from the PP7 affinity purified total RNA samples.
DOI: https://doi.org/10.7554/eLife.33684.015
**Source data 2.** Zipped file contain three files.
DOI: https://doi.org/10.7554/eLife.33684.016
**Source data 3.** FASTA file that contains the windows of enrichment and total gene sequences of genes identified in the PP7 affinity purified total RNA samples.
DOI: https://doi.org/10.7554/eLife.33684.017
**Figure supplement 1.** Identification, purification and biochemical characterization of GsrN-PP7hp chimeras.
DOI: https://doi.org/10.7554/eLife.33684.014

GsrN-PP7hp alleles tagged at the 5' end or at nucleotide positions 54 or 59 did not complement the Δ*gsrN* peroxide survival defect (*Figure 5—figure supplement 1B*). These alleles yielded lower steady-state levels of 5' isoform compared to wild type (*Figure 5—figure supplement 1C,D*). GsrN-PP7hp alleles with insertions at nucleotides 37, 67, and 93 restored peroxide resistance to Δ*gsrN* and produced more 5' isoform than non-complementing GsrN-PP7 constructs (*Figure 5—figure supplement 1*).

The PP7hp aptamer inserted at *gsrN* nucleotide 37 (GsrN(37)::PP7hp) was selected as the bait to identify molecular partners that co-purify with GsrN, as this was the only functional insertion in the 5' half of *gsrN*: as presented above, the 5' end is necessary for function and the 5' isoform is more abundant than the full-length transcript. The pull-down fraction was compared to a negative control pull-down from cells expressing PP7hp fused to the last 50 nucleotides of GsrN including its intrinsic terminator (PP7hp::GsrN-3') (*Figure 5A*). Northern blots demonstrated GsrN-PP7hp fusion transcripts were enriched in our purification (*Figure 5B*). Electrophoretic separation of the eluate followed by silver staining revealed no significant protein differences between GsrN(37)::PP7hp and the negative control (data not shown). Lack of differential protein signal may be due to the conditions in which we performed the pull-down. We identified and quantified co-eluting RNAs by RNA-seq.

We employed two approaches to identify RNAs enriched in GsrN(37)::PP7hp fractions relative to the negative control fractions. A conventional RNA-seq pipeline (*Tjaden, 2015*) quantified mapped reads within annotated gene boundaries as a first pass (*Figure 5C* and *Figure 5—source data 1*). To capture reads in non-coding and unannotated regions, and to analyze reads unevenly distributed across genes, we also developed a sliding window analysis approach. Software we developed to implement Sliding Window Analysis is available on GitHub (*Tien, 2017b*). Specifically, we organized the *Caulobacter* genome into 25 base-pair windows and quantified mapped reads in each window using the EDGE-pro/DESeq pipeline (*Anders and Huber, 2010*; *Magoc et al., 2013*). Together, these two quantification strategies identified several mRNA, sRNAs, and untranslated regions enriched in the GsrN(37)::PP7hp pull-down fraction (*Figure 5D* and *Figure 5—source data 2*). We applied IntaRNA 2.0.2 (*Mann et al., 2017*) to identify potential binding sites between GsrN and the enriched co-purifying RNAs. Of the 67 analyzed enriched genes and regions, 32 of the predicted RNA-RNA interactions involved the cytosine-rich 5' loop in the predicted secondary structure of GsrN (*Figure 5E* and *Figure 5—source data 3*); 31 of targets contained G-rich sequences (*Table 1*). We note that exposed C-rich motifs in sRNAs and G-rich regulatory sequences in mRNA have been reported in several sRNA systems (*Geissmann et al., 2009*; *Papenfort et al., 2008*; *Romilly et al., 2014*; *Sharma et al., 2010*). A sequence logo (*Crooks et al., 2004*) of the predicted target mRNA binding sites is enriched with guanosines (*Figure 5E*), consistent with a model in which six tandem cytosines in the 5' loop of GsrN determine target mRNA recognition. Twenty-seven of the predicted RNA-RNA interactions involved the 3' exposed region of GsrN. The remaining eight enriched genes and regions did not have a significant binding site prediction with GsrN.

Transcripts enriched in the GsrN(37)::PP7hp fraction encode proteins involved in proteolysis during envelope stress, enzymes required for envelope biogenesis, cofactor and nucleotide anabolic enzymes, and transport proteins (*Table 1*). *sigT* and its anti-σ factor, *nepR*, were also enriched in the

**Table 1.** RNAs that co-elute with GsrN-PP7hp.

| Gene locus ID | Gene name | log$_2$ Fold | Identification method | Region(s) | Description | Interacting nucleotides |
|---|---|---|---|---|---|---|
| CCNA_00167 | - | 4.56, 6.95 | Rockhopper, Sliding window | 179311–180120 (+), 179500–179550 (+, I, S) | metallophosphatase family protein | - |
| CCNA_00416 | - | 7.2 | Sliding window | 429625–429725 (-, I, S) | conserved hypothetical membrane protein | GGCGGAGGG |
| CCNA_00587 | - | 4.87 | Sliding window | 616250–616300 (+, I, S) | alpha/beta hydrolase family protein | UCGGCGGGGGGC |
| CCNA_00882 | - | 4.61 | Sliding window | 962875–962925 (-, U, S) | hypothetical protein | UCGGGGGGU |
| CCNA_00894 | - | 4.29 | Sliding window | 974800–974850 (+, I, S) | 1-hydroxy-2-methyl-2-(E)-butenyl 4-diphosphate synthase | UCAAGUCGGGGC |
| CCNA_00897 | - | 3.2 | Rockhopper | 976013–976177 (+) | hypothetical protein | - |
| CCNA_00913 | - | 7.64, 7.80 | Rockhopper, Sliding window | 993033–993209 (-), 993175–993225 (-, I, S) | hypothetical protein | UCAAGUU |
| CCNA_00930 | - | 3.72, 6.98, 3.81 | Rockhopper, Sliding window, Sliding window | 1006253–1006870 (+), 1006275–1006425 (+, I, S), 1006475–1006650 (+, I, S) | riboflavin synthase alpha chain | CGGGUCGGGGGUG |
| CCNA_01024 | - | 3.32 | Rockhopper | 1111617–1112111 (-) | hypothetical protein | CAGGCGGGGGG |
| CCNA_01058 | - | 5.81 | Sliding window | 1159075–1159125 (-, D, S) | helix-turn-helix transcriptional regulator | CGGGGGGU |
| CCNA_01154 | - | 3.45, 6.81 | Rockhopper, Sliding window | 1257902–1258591 (+), 1257975–1258025 (+, I, S) | conserved hypothetical protein | GGGGGCG |
| CCNA_01303 | - | 5.87, 5.30, 8.22 | Rockhopper, Sliding window, Sliding window | 1430061–1430900 (+), 1430550–1430625 (+, I, S), 1430650–1430725 (+, I, S) | conserved hypothetical protein | GGGUCGGGGG |
| CCNA_01304 | - | 2.9 | Rockhopper | 1431129–1431329 (+) | hypothetical protein | GGUUCGCGGACG |
| CCNA_01335 | - | 2.99 | Sliding window | 1448600–1448650 (-, I, S) | ABC-type multidrug transport system, ATPase component | UCGCGUCGA |
| CCNA_01344 | - | 4.62 | Sliding window | 1458550–1458725 (+, I, S) | conserved hypothetical protein | GUCGGGGGUG |
| CCNA_01584 | - | 3.14 | Sliding window | 1699675–1699725 (+, I, A) | multimodular transpeptidase-transglycosylase PBP 1A | GGGGGGC |
| CCNA_01660 | - | 4.41, 6.21 | Rockhopper, Sliding window | 1781219–1781911 (-), 1781350–1781575 (-, I, S) | conserved hypothetical protein | GGGGGCG |
| CCNA_01966 | - | 11.3 | Sliding window | 2110225–2110275 (-, I, A) | vitamin B12-dependent ribonucleotide reductase | GGUCGGGG |
| CCNA_01996 | - | 9.15, 8.86 | Rockhopper, Sliding window | 2142908–2143687 (-), 2143625–2143700 (-, I, S) | undecaprenyl pyrophosphate synthetase | CGGGGGGC |
| CCNA_02034 | - | 7.24 | Sliding window | 2178500–2178550 (+, I, S) | luciferase-like monooxygenase | UCGAUGGGGGGCG |
| CCNA_02064 | lpxC | 3.6 | Sliding window | 2215450–2215550 (-, I, S) | UDP-3-O-(3-hydroxymyristoyl) N-acetylglucosamine deacetylase | UCGGGGGCG |
| CCNA_02089 | - | 8.52 | Rockhopper | 2237967–2238341 (-) | hypothetical protein | UCAAGUCGGGG |
| CCNA_02217 | - | 4.02 | Rockhopper | 2364081–2364383 (-) | hypothetical protein | GCGCGACGAAGG |
| CCNA_02286 | - | 3.26 | Sliding window | 2435450–2435500 (-, I, S) | hypothetical protein | UCCGGUCGCCCGG |
| CCNA_02595 | - | 6.85 | Sliding window | 2743525–2743625 (-, U, S) | Zn finger TFIIB-family transcription factor | UCGCAUCGA |
| CCNA_02758 | - | 2.93 | Rockhopper | 2921763–2922152 (+) | hypothetical protein | UCGCGUC |
| CCNA_02761 | - | 3.65 | Rockhopper | 2923673–2923918 (+) | hypothetical protein | CGGAGGGG |

*Table 1 continued on next page*

*Table 1 continued*

| Gene locus ID | Gene name | log$_2$ Fold | Identification method | Region(s) | Description | Interacting nucleotides |
|---|---|---|---|---|---|---|
| CCNA_02846 | - | 5.44, 8.60 | Sliding window, Sliding window | 3000100–3000175 (-, I, S), 2999225–2999275 (-, I, S) | DegP/HtrA-family serine protease | AAGUCGGGGGGCG |
| CCNA_02860 | - | 3.70, 4.78 | Rockhopper, Sliding window | 3012116–3013060 (-), 3012500–3012550 (-, I, S) | DnaJ-class molecular chaperone | CGGCAAG |
| CCNA_02975 | - | 6.34 | Sliding window | 3130300–3130375 (-, I, A) | excinuclease ABC subunit C | GCGGGGG |
| CCNA_02987 | - | 7.26 | Sliding window | 3142700–3142800 (-, I, A) | hypothetical protein | GUCGGGGGGCGUC |
| CCNA_02997 | cspA | 3.61 | Rockhopper | 3152607–3152816 (-) | cold shock protein CspA | - |
| CCNA_03002 | - | 6.03, 4.48 | Rockhopper, Sliding window | 3155705–3156322 (-), 3155750–3155800 (-, I, S) | CDP-diacylglycerol–glycerol-3-phosphate 3-phosphatidyltransferase | - |
| CCNA_03105 | - | 9.15 | Sliding window | 3255775–3255850 (-, I, S) | DnaJ domain protein | AAGUCGGGGGGUGU |
| CCNA_03113 | - | 3.50, 5.40 | Rockhopper, Sliding window | 3263780–3264499 (-), 3264400–3264450 (-, I, S) | membrane-associated phospholipid phosphatase | UUGUAUCG |
| CCNA_03138 | katG | 3.35 | Sliding window | 3286000–3286050 (+, I, S) | peroxidase/catalase katG | GUCGGGG |
| CCNA_03176 | - | 2.83 | Rockhopper | 3335155–3335445 (-) | nucleotidyltransferase | GAGUUCGCG |
| CCNA_03338 | tolB | 5.27 | Sliding window | 3519425–3519475 (-, I, S) | TolB protein | UCGCGAGGG |
| CCNA_03409 | - | 4.46, 5.44 | Rockhopper, Sliding window | 3576740–3577696 (-), 3577550–3577600 (-, I, S) | alpha/beta hydrolase family protein | GGUUUGUGAAGGG |
| CCNA_03506 | - | 3.27, 4.10 | Rockhopper, Sliding window | 3664090–3664677 (+), 3664100–3664175 (+, I, S) | putative transcriptional regulator | GUUGGGGGG |
| CCNA_03589 | sigT | 3.58 | Rockhopper | 3743953–3744558 (-) | RNA polymerase EcfG family sigma factor sigT | - |
| CCNA_03590 | nepR | 3.43, 3.50 | Rockhopper, Sliding window | 3744561–3744746 (-), 3744675–3744725 (-, I, S) | anti-sigma factor NepR | GGGGGGCG |
| CCNA_03590, CCNA_03589 | nepR-sigT | 4.42 | Sliding window | 3744500–3744575 (-, O, S) | anti-sigma factor NepR, RNA polymerase EcfG family sigma factor sigT | GAGCGUCAACGA |
| CCNA_03617 | - | 3.5 | Rockhopper | 3772262–3772717 (+) | Copper(I)-binding protein | - |
| CCNA_03618, CCNA_03617 | -,- | 6.99 | Sliding window | 3772700–3772750 (+, O, S) | SCO1/SenC family protein, Copper(I)-binding protein | GUCGGGG |
| CCNA_03681 | - | 5.11 | Sliding window | 3843700–3843750 (-, U, S) | ABC transporter ATP-binding protein | UCAGUUGGGG |
| CCNA_03825 | - | 3.63 | Rockhopper | 3991412–3991774 (-) | hypothetical protein | GGGGGCGU |
| CCNA_03825, CCNA_03826 | -,- | 8.01 | Sliding window | 3991750–3991825 (-, O, S) | hypothetical protein, conserved hypothetical protein | GGGGGCGU |
| CCNA_03826 | - | 3.71 | Rockhopper | 3991771–3992325 (-) | conserved hypothetical protein | - |
| CCNA_03888 | - | 2.99 | Rockhopper | 761965–762324 (+) | conserved hypothetical protein | GCGGUCCGG |
| CCNA_03976 | - | 3.57 | Rockhopper | 2923462–2923683 (+) | hypothetical protein | GAGCGCGUCGGCA |
| CCNA_R0016 | - | 8.53 | Rockhopper | 844332–844401 (+) | small non-coding RNA | UCGGGGG |
| CCNA_R0035 | - | 6.64 | Rockhopper | 1549367–1549443 (+) | tRNA-Pro | AAGGGGU |
| CCNA_R0044 | - | 4.82 | Rockhopper | 2059848–2059942 (-) | complex medium expressed sRNA | - |
| CCNA_R0061 | - | 4.65 | Sliding window | 2800475–2800525 (-, I, S) | RNase P RNA | UAGGUCGGGGC |
| CCNA_R0089 | - | 3.3 | Sliding window | 3874375–3874425 (+, U, S) | tRNA-Ala | UCGGGGGGCG |
| CCNA_R0100 | - | 4.4 | Rockhopper | 165492–165575 (+) | small non-coding RNA | CGGAGGG |
| CCNA_R0108 | - | 4.27 | Rockhopper | 472905–472973 (+) | small non-coding RNA | UCGGGGG |
| CCNA_R0180 | - | 5.14 | Rockhopper | 3266851–3266937 (-) | small non-coding RNA | - |

Gene Locus ID: GenBank locus ID
Gene Name: if available log$_2$Fold: calculated fold change of the given region
Identification Method: refers to what strategy identified the enriched gene in the PP7hp affinity purification RNA-Seq
Region(s): the region and strand used to calculate the log$_2$Fold metric. Additionally for the sliding window analysis additional information is provided. First letter indicates the relative position of the region indicated to the annotated gene coordinates: I-internal, U-upstream, D-downstream. Second letter indicates the direction in which the reads mapped: S-sense, A-anti-sense.
Description: product description of the given gene(s)
Interacting nucleotides: the nucleotides within the proposed chromosomal region(s) that are predicted to interact with GsrN.
DOI: https://doi.org/10.7554/eLife.33684.018

GsrN(37)::PP7hp fraction, though we found no evidence for regulation of σ$^T$/NepR by GsrN (*Figure 1—figure supplement 2D*). We observed significant enrichment of rRNA in the GsrN(37)::PP7hp fractions; the functional significance of this signal is not known (grey points above and to the right of the red cut-off *Figure 5D*). *katG*, which encodes the sole catalase-peroxidase in the *Caulobacter* genome (*Marks et al., 2010*), was among the highly enriched mRNAs in our pull-down. Specifically, reads mapping to the first 60 nucleotides of *katG* including the 5' leader sequence and the first several codons of the open-reading frame were enriched in the GsrN(37)::PP7hp pull-down fraction relative to the negative control (*Figure 5F*). *katG* was an attractive GsrN target to interrogate the mechanism by which GsrN determines cell survival under hydrogen peroxide stress.

## GsrN base pairing to the 5' leader of *katG* activates *katG* translation, and enhances peroxide stress survival

Most bacterial sRNAs regulate gene expression at the transcript and/or protein levels through Watson-Crick base pairing with the 5'end of their mRNA targets (*Wagner and Romby, 2015*). We sought to test whether GsrN affected the expression of *katG*. GsrN did not effect *katG* transcription in exponential or stationary phases, or in the presence of peroxide as measured by a *katG-lacZ* transcriptional fusion (*Figure 6—figure supplement 1A–C*). However, *katG* is transcriptionally regulated by the activator OxyR, which binds upstream of the predicted −35 site in the *katG* promoter (*Italiani et al., 2011*). To decouple the effects of OxyR and GsrN on *katG* expression, we generated a strict *katG* translational reporter that contains the mRNA leader of *katG* fused in-frame to *lacZ* (*katG-lacZ*) constitutively expressed from a σ$^{RpoD}$-dependent promoter. In both exponential and stationary phases, *katG-lacZ* activity is reduced in Δ*gsrN* and enhanced in *gsrN*$^{++}$ strains compared to wild type (*Figure 6—figure supplement 1D,F*). Hydrogen peroxide exposure did not affect *katG-lacZ* activity (*Figure 6—figure supplement 1E*). We conclude that GsrN enhances KatG protein expression, but not *katG* transcription.

We then used this translational reporter to investigate a predicted binding interaction between the unpaired 5' loop of GsrN and a G-rich region at the 5' end of the *katG* transcript. Specifically, the first 7 nucleotides of *katG* mRNA (*Zhou et al., 2015*) are complementary to seven nucleotides in the single-stranded 5' loop of GsrN, including 4 of the six cytosines (*Figure 6A*). We disrupted this predicted base pairing, mutating 5 of the seven nucleotides in the putative *katG* target site and GsrN interaction loop. These mutations preserved GC-content, but reversed and swapped (RS) the interacting nucleotides (*Figure 6A*). We predicted that pairs of wild-type and RS mutant transcripts would not interact, while base pairing interactions would be restored between RS mutant pairs.

Mutating the predicted target site in the *katG* 5' leader ablated GsrN-dependent regulation of the *katG-lacZ* translational reporter (*Figure 6—figure supplement 2A*); expression was reduced to a level similar to Δ*gsrN*. We further tested this interaction by assessing the effect of the reverse-swapped *gsrN*(RS) allele on the expression of *katG-lacZ*. However, GsrN(RS) was unstable; total GsrN(RS) levels were ≈10-fold lower than wild-type GsrN (*Figure 6—figure supplement 3A,C*). To overcome GsrN(RS) instability, we inserted a plasmid with three tandem copies of *gsrN*(RS), 3*gsrN*(RS), into the *vanA* locus in a Δ*gsrN* background, which increased steady-state levels of GsrN(RS) approximately 4-fold (*Figure 6—figure supplement 3A,C*). *katG* target site or GsrN recognition loop mutations significantly reduced *katG-lacZ* expression (Student's t-test, p=0.0026 and p=0.0046, respectively). Compensatory RS mutations that restored base pairing between the *katG* target site and the GsrN loop rescued *katG-lacZ* expression (*Figure 6B*).

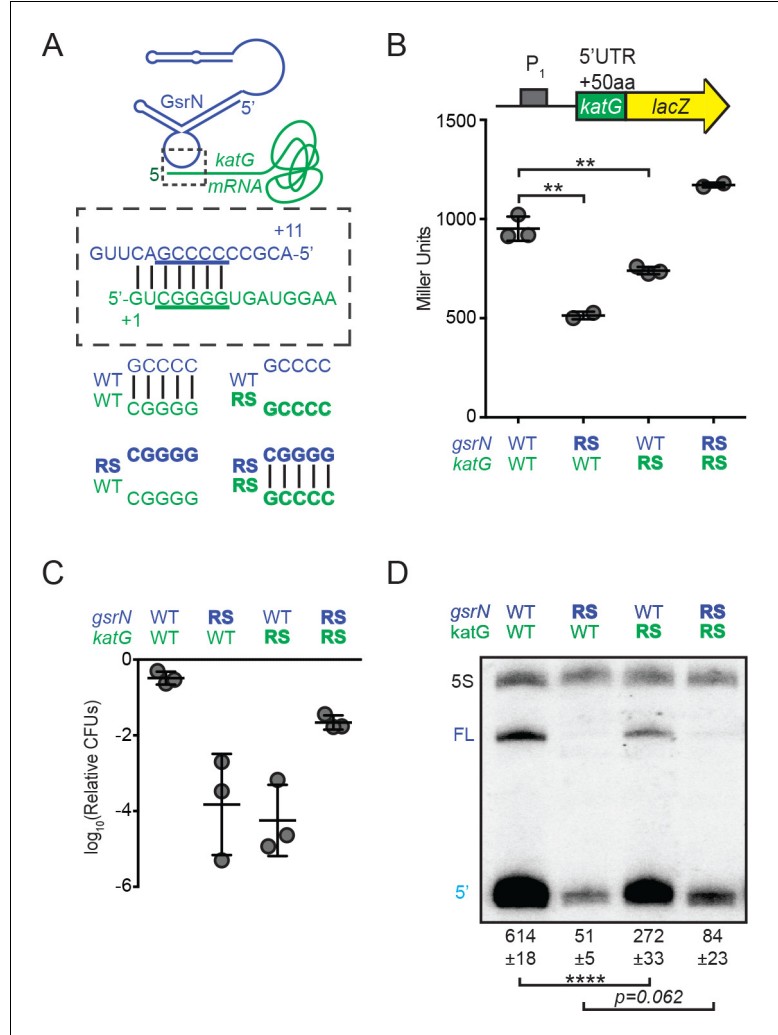

**Figure 6.** GsrN base pairs with the 5' leader of *katG* mRNA and enhances KatG expression. (**A**) Predicted interaction between GsrN (blue) and *katG* mRNA (green), with base-pairing shown in dashed box. Wild-type (WT) and reverse-swapped (RS) mutation combinations of the underlined bases are outlined below. (**B**) Translation from *katG* and *katG-RS* reporters in Δ*gsrN* strains expressing 3*gsrN* (WT) or 3*gsrN(RS)* (RS). Measurements were taken from exponential phase cultures. Bars represent mean ±SD of at least two independent cultures (points). ** p-value<0.01 estimated by Student's t-test. (**C**) Relative hydrogen peroxide survival of RS strains. Δ*gsrN* strains expressing 3*gsrN* or 3*gsrN(RS)* and encoding *katG* or *katG(RS)* alleles. Bars represent mean ±SD from three independent experiments (points). (**D**) Northern blot of total RNA from strains in (**C**) collected in exponential phase hybridized with probes complementary to 5' end of GsrN and 5S rRNA. Quantification is mean ±SD normalized signal from three independent experiments. **** p-value<0.0001 estimated by Student's t-test.
DOI: https://doi.org/10.7554/eLife.33684.019

The following figure supplements are available for figure 6:

**Figure supplement 1.** *gsrN* does not regulate *katG* transcription, but does enhance *katG-lacZ* mRNA translation.
DOI: https://doi.org/10.7554/eLife.33684.020

**Figure supplement 2.** *katG(RS)-lacZ* translation is not affected by *gsrN*.
DOI: https://doi.org/10.7554/eLife.33684.021

**Figure supplement 3.** *gsrN* levels are determined by the sequence of its target recognition loop and *gsrN(RS)* allele cannot complement the peroxide susceptibility of Δ*gsrN*.
DOI: https://doi.org/10.7554/eLife.33684.022

To assess the physiological consequence of mutating the G-tract in the *katG* mRNA leader and the GsrN C-rich loop, we replaced wild-type *katG* on the chromosome with the *katG(RS)* allele in both the Δ*gsrN* +3 gsrN and Δ*gsrN* +3*gsrN*(RS) backgrounds, and measured survival after hydrogen peroxide exposure. Both *katG(RS)* and *gsrN(RS)* mutants had survival defects (*Figure 6C* and *Figure 6—figure supplement 3B*). Strains harboring the *katG(RS)* allele phenocopy Δ*gsrN* under peroxide stress. While *katG(RS)* survival is compromised, the defect is not as large as a strain missing *katG* completely (Δ*katG*) (compare *Figure 6C* and *Figure 6—figure supplement 2C*). Expressing *gsrN* (RS) in one, three, or six tandem copies did not complement the peroxide survival defect of Δ*gsrN* (*Figure 6—figure supplement 3B*). The peroxide survival defect of the individual RS alleles is restored in the strain carrying both *katG(RS) and gsrN(RS)* alleles, which has restored base pairing between the GsrN 5' loop and the *katG* 5' leader (*Figure 6C*). We conclude that base paring between the *katG* leader and the GsrN loop is critical for *katG* expression and peroxide stress survival.

We note that the protective effect of *gsrN* overexpression is lost when *katG* is deleted. Moreover, the peroxide survival defect of Δ*gsrN* cells can be rescued by overexpression of *katG* (*Figure 6—figure supplement 2C*). We conclude that *katG* is necessary and sufficient to protect the cell from hydrogen peroxide and that GsrN modulates expression of *katG*.

Given differences in steady state levels of GsrN and GsrN(RS), we postulated that the capacity of GsrN to interact with its targets influences its stability in vivo. Indeed, mutation of the *katG* target site reduced GsrN by more than two-fold (Student's t-test, p<0.0001). The compensatory *katG(RS)* allele partially restored stability to GsrN(RS) (*Figure 6D*). *katG(RS)* mutation or *katG* deletion did not influence *gsrN* transcription (*Figure 6—figure supplement 2B*). Thus, we attribute the differences in steady-state levels of the GsrN alleles to their ability to interact with mRNA targets via the 5' C-rich loop.

## GsrN enhances KatG expression and stabilizes *katG* mRNA in vivo in the presence of peroxide

To assess the relative effects of GsrN on *katG* transcript and protein levels in vivo, we directly measured both by dot blot and Western blot, respectively. In untreated and peroxide treated cultures, *katG* transcript levels trended lower in Δ*gsrN* and higher in *gsrN*[++] compared to wild type (*Figure 7A*). In untreated cultures, these transcript differences are not statistically significant (Student's t-test, p=0.39) yet KatG protein tagged with the M2 epitope was reduced two-fold in Δ*gsrN* lysate relative to wild-type (Student's t-test, p<0.0001) (*Figure 7*). Upon peroxide treatment, steady-state *katG* transcript levels differ significantly between Δ*gsrN* and *gsrN*[++] cultures (Student's t-test, p<0.01) (*Figure 7A*). KatG-M2 protein was reduced 3-fold in Δ*gsrN* lysate relative to wild-type, and overexpression of *gsrN* increased KatG-M2 two-fold compared to wild-type (*Figure 7B*). Since GsrN does not influence *katG* transcription (*Figure 6—figure supplement 1A,C*), these data support a model in which GsrN enhances KatG translation in vivo.

Peroxide treatment results in approximately five-fold induction of *katG* mRNA in both wild-type and *gsrN* mutant strains. We attribute this to OxyR-dependent activation of *katG*, independent of *gsrN*. The corresponding induction of KatG protein is modest (1.5 to 2-fold) in wild-type and *gsrN*[++] strains after 15 min of peroxide treatment. Given the 15 min treatment period, this discrepancy in fold-change may reflect faster accumulation of transcript than protein and/or inefficient *katG* translation. In Δ*gsrN* cells, *katG* transcript is induced by peroxide yet KatG protein does not change significantly. Thus, despite OxyR-induced transcription, efficient translation of *katG* mRNA requires GsrN.

## GsrN is a general regulator of stress adaptation

In the GsrN::PP7hp pull-down fraction, multiple RNAs in addition to *katG* were enriched (*Figure 5C, D*). This suggested that GsrN may have regulatory roles beyond mitigation of peroxide stress. To globally define genes that are directly or indirectly regulated by GsrN, we performed RNA-seq and LC-MS/MS measurements on wild-type, Δ*gsrN* and *gsrN*[++] strains (*Figure 8A* and *Figure 8—source data 1*). We identified 40 transcripts, including *gsrN*, with significant differences in mapped reads between the Δ*gsrN* and *gsrN*[++] samples (*Figure 8—figure supplement 1A* and *Figure 8—source data 2*). Eleven proteins had significant label free quantitation (LFQ) differences (FDR < 0.05) between *gsrN*[++] and Δ*gsrN* (*Figure 8—figure supplement 1B* and *Figure 8—source data 3*). Most

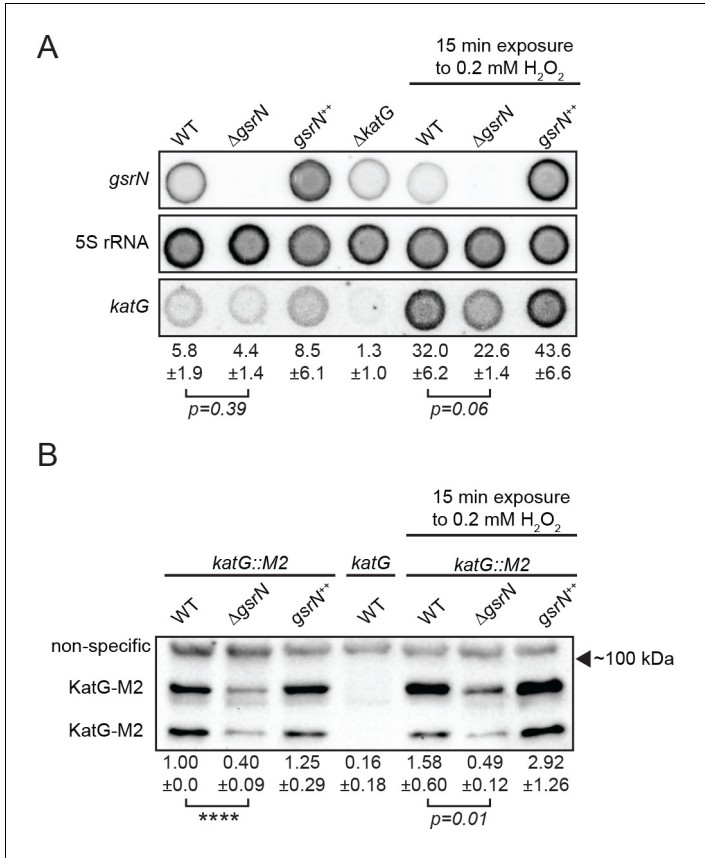

**Figure 7.** GsrN affects KatG and *katG* mRNA levels in vivo. (**A**) Dot blot of total RNA of *gsrN* and *katG* mutants grown to early stationary phase ($OD_{660}$0.85–0.9; this is the growth phase we used to initiate stress assays). Samples on right were treated with 0.2 mM hydrogen peroxide for 15 min before RNA extraction. These conditions were chosen to evaluate the effects of peroxide without ablating the $\Delta gsrN$ cultures. Blots were hybridized with *katG* mRNA, GsrN or 5S rRNA probes. *katG* mRNA signal normalized to 5S rRNA signal is quantified (mean ±SD, n = 3, p-value estimated with Student's t-test). (**B**) Immunoblot of KatG-M2 fusion in wild type, $\Delta gsrN$, and $gsrN^{++}$ strains in the presence and absence of peroxide stress probed with α-FLAG antibody. KatG migrates as two bands as previously reported (*Italiani et al., 2011*). Normalized KatG-M2 signal (mean ±SD, n = 4, ****p<0.0001 Student's t-test) is presented below each lane. Arrow indicates position of 100 kDa molecular weight marker.
DOI: https://doi.org/10.7554/eLife.33684.023

genes identified as significantly regulated by transcriptomic and proteomic approaches did not overlap. This is not surprising, and would be expected for cases like *katG* where GsrN modulates translation, but not transcription.

We note there is little overlap between the transcripts enriched in the GsrN(37)-PP7hp pull-down fraction, and proteins identified as significantly regulated in our global proteomic measurements. This may be partially due to limited coverage: our protein mass spectrometry data only captured 30% of the annotated proteome. Nonetheless, these data provide evidence that GsrN can function as both a positive and negative regulator of gene expression, either directly or indirectly.

Importantly, RNA-seq and proteomics experiments validated *katG* as a regulatory target of GsrN. *katG* transcript levels measured by RNA-seq were not significantly different between $\Delta gsrN$ and $gsrN^{++}$ strains (*Figure 8B*), consistent with our dot blot measurements of unstressed cultures (*Figure 7A*). Conversely, steady-state KatG protein levels estimated from our LC-MS/MS experiments were significantly reduced in $\Delta gsrN$, consistent with our western blot analysis of KatG protein (*Figures 8C* and *7B*). *katG* was the only gene that was significantly enriched in the pull-down and differentially expressed in the proteomic studies (*Figure 8A*). These results provide additional evidence that *katG* transcript is a major target of GsrN, and that GsrN functions to enhance KatG expression at the post-transcriptional level.

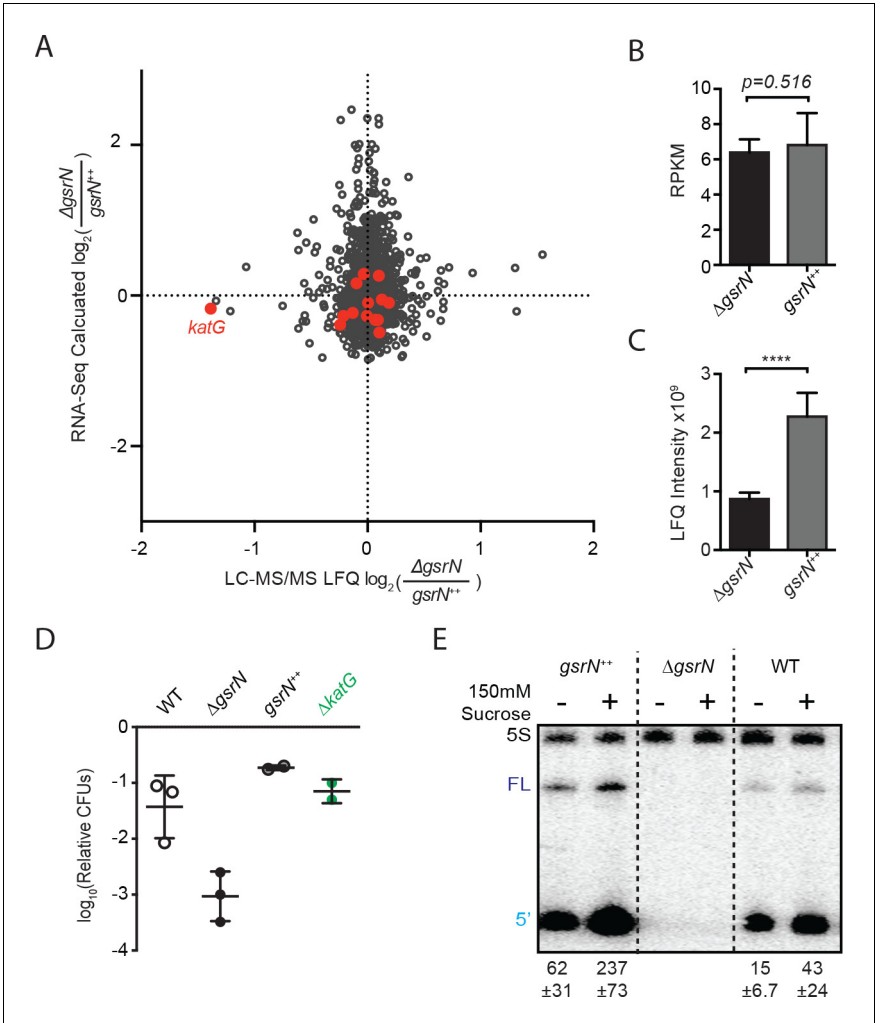

**Figure 8.** GsrN is a global regulator of stress physiology. (**A**) Transcriptomic and proteomic analysis of Δ*gsrN* and *gsrN*[++] strains in early stationary phase (***Figure 8—source data 1***). Only genes detected in both analyses are plotted. Red indicates transcripts that co-purify with GsrN-PP7hp (***Figure 5C,D***). (**B**) *katG* transcript from Δ*gsrN* and *gsrN*[++] cells quantified as reads per kilobase per million mapped (RPKM). Data represent mean ±SD of five independent samples. Significance was evaluated with the Wald test. (**C**) Label free quantification (LFQ) intensities of KatG peptides from Δ*gsrN* and *gsrN*[++] cells (mean ±SD, n = 3; ****p<0.0001 Student's t-test). (**D**) Hyperosmotic stress survival of wild type, Δ*gsrN*, *gsrN*[++], and Δ*katG* cells relative to untreated cells. Stress was a 5 hr treatment with 300 mM sucrose. These conditions were chosen to highlight the dynamic range between Δ*gsrN* susceptibility and *gsrN*[++] protection. Data represent mean ± SD from two independent experiments (points). (**E**) Northern blot of total RNA from wild type, Δ*gsrN*, and *gsrN*[++] cultures with or without 150 mM sucrose stress. Blots were hybridized with GsrN and 5S rRNA probes. Normalized mean ± SD of total GsrN signal from three independent samples is quantified.

DOI: https://doi.org/10.7554/eLife.33684.024

The following source data and figure supplements are available for figure 8:

**Source data 1.** Excel file that contains the log2Fold calculated values from both LC-MS/MS and RNA-Seq analysis of Δ*gsrN* versus *gsrN*[++].

DOI: https://doi.org/10.7554/eLife.33684.027

**Source data 2.** Excel file that contains the compiled information from the CLC workbench analysis.

DOI: https://doi.org/10.7554/eLife.33684.028

**Source data 3.** Excel file that contains the multiple t-test analysis outlined in Materials and methods- LC-MS/MS processing of total soluble protein.

DOI: https://doi.org/10.7554/eLife.33684.029

**Figure supplement 1.** GsrN directly or indirectly affects the expression of multiple genes.

*Figure 8 continued on next page*

*Figure 8 continued*

DOI: https://doi.org/10.7554/eLife.33684.025

**Figure supplement 2.** GsrN is required for osmotic stress survival.

DOI: https://doi.org/10.7554/eLife.33684.026

Given our transcriptomic and proteomic datasets, we reasoned that GsrN may contribute to other phenotypes associated with deletion of the GSR sigma factor, *sigT*. Indeed, the Δ*gsrN* mutant exhibits a survival defect after exposure to hyperosmotic stress, similar to Δ*sigT*, while *gsrN* overexpression protects cells under this condition (*Figure 8D*). Hyperosmotic stress survival does not require *katG* (*Figure 8D*), providing evidence that a distinct GsrN regulatory target mediates this response. Unlike hydrogen peroxide (*Figure 7A*), hyperosmotic stress induces GsrN expression (*Figure 8E*). This is consistent with previous transcriptomic studies in *Caulobacter* in which hyperosmotic stress, but not peroxide stress, activated GSR transcription (*Alvarez-Martinez et al., 2007*). GsrN transcription is also significantly enhanced in stationary phase cultures relative to logarithmic phase cultures (*Figure 1—figure supplement 2E*). Although its functional role under this condition remains undefined, it has been reported that *katG* is a genetic determinant of stationary phase survival (*Steinman et al., 1997*).

## σ$^{EcfG}$-regulated sRNAs are prevalent across the alphaproteobacterial clade

The GSR system is broadly conserved in Alphaproteobacteria. Given the importance of GsrN as a post-transcriptional regulator of the *Caulobacter* GSR, we reasoned that functionally-related sRNAs might be a conserved feature of the GSR in this clade. To identify potential orthologs of *gsrN*, we surveyed the genomes of Alphaproteobacteria that encoded regulatory components of the GSR system (*phyR*, ecfG/*sigT* and *nepR* homologs) and for which transcriptomic data were publically available.

We initially searched for GsrN-related sequences using BLASTn (*Altschul et al., 1990*). Hits to GsrN were limited to the Caulobacteraceae family, including the genera *Caulobacter*, *Brevundimonas*, and *Phenylobacterium*. The 5′ C-rich loop of homologs identified in this family had the highest level of conservation compared to other regions of secondary structure (*Figure 9B*). Predicted *gsrN* homologs are typically proximal to the genes encoding the core GSR regulators (*ecfG/sigT*, *nepR* and *phyR*) (*Figure 9A*). *C. crescentus* is a notable exception where *gsrN* is positioned distal to the GSR locus. Therefore, we used genome position as a key parameter to identify additional GsrN or GsrN-like RNAs in Alphaproteobacteria outside of Caulobacteraceae.

Our search for GsrN focused on three parameters: evidence of intergenic transcription, identification of a near-consensus σ$^{EcfG}$-binding site in the promoter region, and proximity to the *sigT-phyR* chromosomal locus. Based on our criteria, we identified a set of putative GsrN homologs in the Rhizobiaceae family (*Jans et al., 2013*; *Kim et al., 2014*; *Valverde et al., 2008*) (*Figure 9A*). The predicted secondary structure of these putative GsrN homologues has features similar to GsrN from Caulobacteraceae. Specifically, there is an exposed cytosine-rich loop at the 5′ end (*Figure 9C*). This analysis predicts that GsrN-related small RNAs have a functional role in regulating the general stress response in related Alphaproteobacteria.

## Discussion

We sought to understand how GSR transcription determines cell survival across a spectrum of chemical and physical conditions. To this end, we developed a directed gene network analysis approach to predict genes with significant functional roles in the *Caulobacter* GSR. Our approach led to the discovery of *gsrN*, a small RNA of previously unknown function that is a major post-transcriptional regulator of stress physiology.

### Role of GsrN in mitigating hydrogen peroxide stress

Hydrogen peroxide can arise naturally from the environment and is also produced as an aerobic metabolic byproduct (*Imlay, 2013*). Our data provide evidence that σ$^T$-dependent transcription of GsrN basally protects cells from hydrogen peroxide by enhancing KatG expression. Unlike the

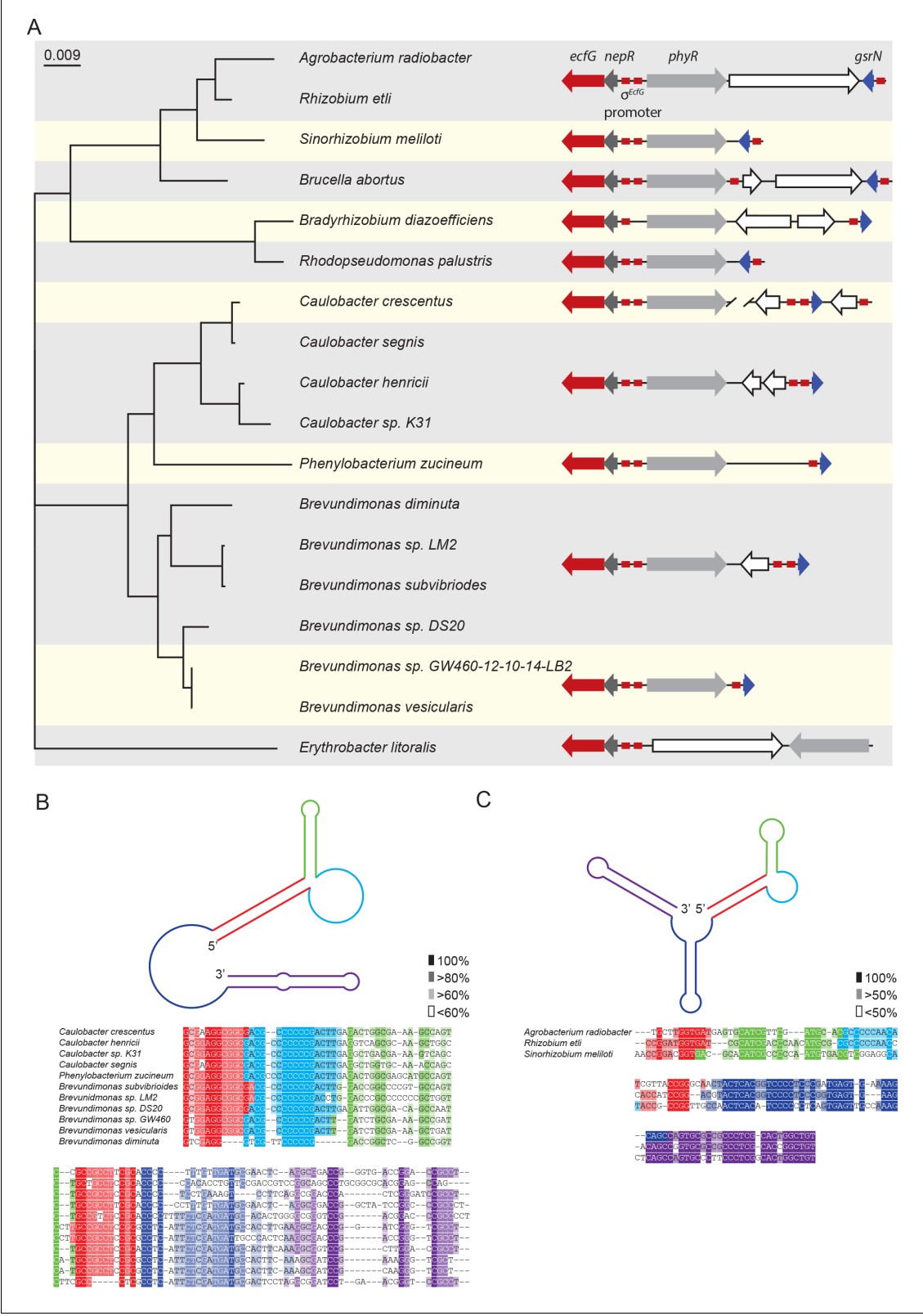

**Figure 9.** Conserved features of GsrN homologues. (**A**) Locus diagrams showing predicted *gsrN* homologs in several Alphaproteobacteria. Tree was constructed from the 16 s rRNA sequences of each strain where *Erythrobacter litoralis* (for which there is no apparent *gsrN*-like gene) was the out-group. Red arrows represent *ecfG*, dark gray arrows represent *nepR*, light gray arrows represent *phyR*, and dark blue arrows represent *gsrN* (or its putative homologs). Red boxes represent the conserved σ^ecfG^-binding site. The prediction of GsrN orthologs in the Caulobacteraceae (*Caulobacter*, *Figure 9 continued on next page*

*Figure 9 continued*

*Brevundimonas*, and *Phenylobacterium*) was based on a BLASTn search (*Altschul et al., 1990*). The prediction of GsrN in *Rhizobium etli*, *Sinorhizobium meliloti*, and *Brucella abortus* was based on evidence of GSR-dependent expression in published transcriptome data, proximity to the GSR locus, and identification of a $\sigma^{ecfG}$-binding site upstream of the gene. The prediction of *Agrobacterium radiobacter* was based on a BLASTn search of using the predicted GsrN sequence from *R. etli* as the query (*Altschul et al., 1990*). The prediction of *Rhodopseudomonas palustris* and *Bradyrhizobium diazoefficiens* is solely based on the proximity to the GSR locus and the presence of an upstream $\sigma^{ecfG}$-binding site. (B) Diagram of predicted secondary structure of GsrN in other Caulobacteraceae is colored by secondary structure element. Colors highlighted in the sequence alignment correspond to the predicted secondary structure regions in the cartoon. Density of shading corresponds to conservation at that position. (C) Diagram of predicted secondary structure of predicted GsrN homologs in select Rhizobiaceae where the 5' portion contains an unpaired 5' G-rich loop (cyan) flanked by a small hairpin (green) and a stem loop involving the 5' terminus (red).

DOI: https://doi.org/10.7554/eLife.33684.030

transcription factor OxyR, which induces *katG* expression in response to peroxide (*Italiani et al., 2011*), GsrN is not induced by peroxide treatment. KatG levels change by only a factor of two when *gsrN* is deleted or overexpressed, but we observe large peroxide susceptibility and protection phenotypes as a function of *gsrN* deletion and overexpression, respectively. The survival phenotypes associated with subtle fold changes in KatG expression suggest that the capacity of KatG to detoxify endogenous sources of $H_2O_2$ is at or near its limit under normal cultivation conditions, similar to what has been postulated for *E. coli* (*Imlay, 2013*).

Expression of the ferritin-like protein, Dps, is controlled by $\sigma^T$ and is reported to aid in the survival of *Caulobacter* under peroxide stress (*de Castro Ferreira et al., 2016*). The protective effect of Dps is apparently minimal under our conditions given that *a)* the peroxide survival defect of $\Delta sigT$ is rescued by simply restoring *gsrN* transcription (*Figure 2B,C*), and *b)* survival after peroxide exposure is determined almost entirely by modifying base-pairing interactions between GsrN and *katG* mRNA. This stated, the difference in hydrogen peroxide susceptibility between $\Delta sigT$ and $\Delta gsrN$ (*Figure 1E*) may be explained in part by the fact that *dps* is still expressed in $\Delta gsrN$ cells.

## Post-transcriptional gene regulation by GsrN is a central feature of the general stress response

Our data define *Caulobacter* GsrN as a central regulator of stress physiology that *1)* is transcribed by the general stress response (GSR) sigma factor ($\sigma^T$) and *2)* has a major protective effect across distinct conditions. Multiple sRNA regulators of bacterial GSR systems have been described. In the case of *E. coli*, *rpoS* translation is controlled by sRNAs, DsrA, RprA, and ArcZ, which are necessary for survival under acid stress (*Bak et al., 2014*; *Pernitzsch et al., 2014*). In the case of GsrN, we find no evidence for feedback on $\sigma^T$ expression and activity, although *nepR-sigT* mRNA did co-elute with GsrN in a pull-down (*Table 1* and *Figure 1—figure supplement 2D*). The regulatory effects we observe are, apparently, purely post-transcriptional and downstream of $\sigma^T$. In this way, GsrN functions like a number of other sRNA regulated downstream of $\sigma^S$ and $\sigma^B$ of Gammaproteobacteria and Firmicutes, respectively (*Fröhlich et al., 2016*; *Fröhlich and Vogel, 2009*; *Mäder et al., 2016*; *Mellin and Cossart, 2012*; *Opdyke et al., 2004*; *Romilly et al., 2014*; *Silva et al., 2013*). GsrN protects *Caulobacter* from hyperosmotic and peroxide stress conditions via genetically distinct post-transcriptional mechanisms (*Figures 1* and *8*). We conclude that transcriptional activation of GsrN by $\sigma^T$ initiates a downstream post-transcriptional program that directly affects multiple genes required for stress mitigation (*Figure 10*).

Quantitative proteomic studies (*Figure 8A*) demonstrate that GsrN activates and represses protein expression, either directly or indirectly. In the case of KatG, we have shown that GsrN is among the rare class of sRNAs that directly enhance protein expression (*Papenfort and Vanderpool, 2015*) (*Figures 7B* and *8C*). Our global and directed measurements of mRNA show that *katG* mRNA levels do not change significantly between $\Delta gsrN$ and $gsrN^{++}$ strains (*Figures 7* and *8*). However, in the presence of peroxide, we observe significant changes in *katG* mRNA that correlate with changes in KatG protein levels. Our data suggest a role for GsrN as a regulator of *katG* translation and, perhaps, *katG* mRNA stability. In this way, GsrN may be similar to the sRNAs, DsrA and RyhB (*Lease and Belfort, 2000*; *Prévost et al., 2007*), which function by uncovering ribosome-binding sites (RBS) in the leaders of their respective mRNA targets. Although this mechanism may occur for GsrN-*katG*, we are not able to predict a clear RBS in the *katG* mRNA leader, which is among the

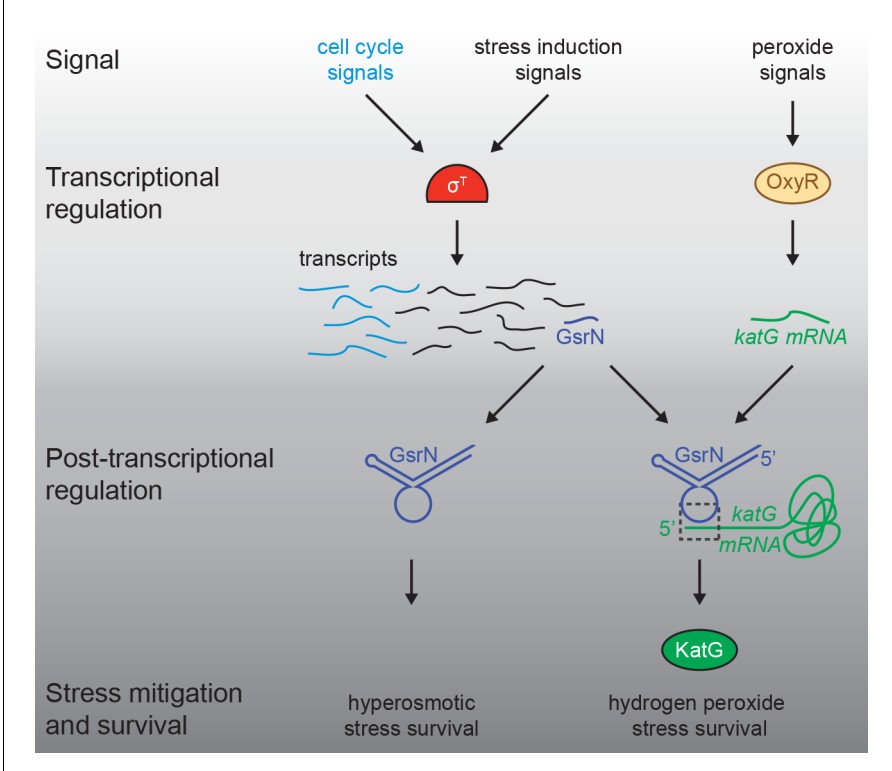

**Figure 10.** Regulatory architecture of the *Caulobacter* stress response systems. Expression of the GSR EcfG-sigma factor, *sigT* (σ[T]), and select genes in the GSR regulon is regulated as a function cell cycle phase. σ[T]-dependent transcription can be induced by certain signals (e.g. hyperosmotic stress), but is unaffected by hydrogen peroxide. Transcription of the sRNA, GsrN, is activated by σ[T], and the cell cycle expression profile of *gsrN* is highly correlated with *sigT* and its upstream regulators. Transcription of the catalase/peroxidase *katG* is independent of σ[T]. GsrN dependent activation of KatG protein expression is sufficient to rescue the peroxide survival defect of a Δ*sigT* null strain. GsrN convenes a post-transcriptional layer of gene regulation that confers resistance to peroxide and hyperosmotic stresses.

DOI: https://doi.org/10.7554/eLife.33684.031

75% of open-reading frames (ORFs) in *Caulobacter* that do not contain a canonical RBS (*Schrader et al., 2014*). GsrN binding to the 5' end of *katG* mRNA could recruit the ribosome to a potential stand-by binding site. Alternatively, it could induce a structural change that enhances ribosome binding, which is a mechanism proposed for the sRNA, RepG (*Pernitzsch et al., 2014*). Steady state levels of *katG* mRNA are influenced by GsrN, which may be a result of sRNA-mRNA binding (*Fröhlich et al., 2013*; *Ramirez-Peña et al., 2010*) or GsrN-dependent mRNA processing (*Obana et al., 2010*).

GsrN is a remarkable sRNA that has at least two distinct physiological roles in the cell, mitigating peroxide stress and hyperosmotic stress. The target of GsrN that confers hyperosmotic stress protection remains undefined, but this phenotype is also likely regulated at the post-transcriptional level (*Figure 8D*). While the reported physiological effects of sRNAs are often subtle, GsrN provides a notable example of a single post-transcriptional regulator that exerts a strong influence on multiple, distinct pathways affecting cellular stress survival.

## On GsrN stability and processing

The roles of sRNAs in stress adaptation have been investigated in many species, and a number of molecular mechanisms underlying sRNA-dependent gene regulation have been described. We have uncovered a connection between mRNA target site recognition and GsrN stability that presents challenges in the characterization of GsrN regulatory mechanisms. Specifically, mutations in the *katG* mRNA leader affect steady-state levels of GsrN (*Figure 6D*). Given this result, one could envision

scenarios in which changes in transcription of *katG* or some other direct GsrN target could broadly affect stress susceptibility by altering levels of GsrN and, in turn, the stability of other target mRNAs in the cell. In short, the concentrations of mRNA targets could affect each other via GsrN. Such effects should be considered when assessing mRNA target site mutations in this system and others.

GsrN is among a group of sRNAs that are post-transcriptionally processed (*Chao et al., 2017*; *Chao et al., 2012*; *Mandin and Gottesman, 2010*; *Papenfort et al., 2015*; *Papenfort et al., 2009*) (*Figure 3*). Other examples of processed sRNAs include the *rpoS* regulator, RprA, for which multiple isoforms have been observed. Unlike GsrN, *rprA* yields a stable 3' isoform that arises from endonucleolytic-cleavage (*Papenfort et al., 2009*). The 3' isoform of RprA does not apparently influence *rpoS* expression, but regulates a subset of mRNAs via a second identified base-pairing region (*Papenfort et al., 2015*). Like RprA, the distinct GsrN isoforms may have different target preferences.

Select PP7hp insertions resulted in reduced 5' isoform formation. PP7hp mutants with low 5' isoform levels did not complement the peroxide viability defect of Δ*gsrN*. Processing to a short 5' isoform may thus be necessary for GsrN to bind *katG* mRNA and regulate KatG expression. Alternatively, processing may not be required for function, and lack of complementation by certain hairpin insertion mutants could be due to PP7hp interfering with target recognition or simply reducing total levels of GsrN. Regardless, our data clearly show that GsrN is cleaved to yield a 5' isoform that is stable in the cell (*Figure 3—figure supplement 1*) and is sufficient to protect *Caulobacter* from hydrogen peroxide treatment (*Figure 4B*). The role of RNA metabolism in sRNA-dependent gene regulation is not well understood. GsrN will likely provide a good model to investigate mechanisms by which mRNA target levels and sRNA/mRNA processing control gene expression.

### *Caulobacter* GSR and the cell cycle

The transcription of *sigT*, *gsrN*, and several other genes in the GSR regulon are cell cycle regulated (*Fang et al., 2013*; *Laub et al., 2000*; *McGrath et al., 2007*; *Zhou et al., 2015*), with highest expression during the swarmer-to-stalked cell transition, when cells initiate DNA replication and growth (*Figure 1C*). GSR activation during this period potentially protects cells from endogenous stressors that arise from upregulation of anabolic systems required for growth and replication. In the future, it is of interest to explore the hypothesis that the GSR system and GsrN provide both basal protection against endogenous stressors generated as a function of normal metabolism and induced protection against particular stressors (e.g. hyperosmotic stress) encountered in the external environment.

## Materials and methods

**Key resources table**

| Reagent type | Designation | Source or reference | Identifier |
| --- | --- | --- | --- |
| Antibody | Goat anti-Mouse IgG (H + L) Secondary Antibody, HRP | ThermoFisher | 32430 |
| Antibody | DYKDDDDK Tag Monoclonal Antibody (FG4R) | ThermoFisher | MA1-91878-1MG |
| Strain, strain background | See *Supplementary file 1* | | |
| Chemical compound, drug | Agar | Lab Scientific | A466 |
| Chemical compound, drug | 30% Hydrogen Peroxide | ThermoFisher | H325-100 |
| Chemical compound, drug | substrate o-nitrophenyl-β-D-galactopyranoside (ONPG) | GoldBio | N-275–100 |
| Chemical compound, drug | acrylamide:bisacrylamide (29:1) | BioRad | 1610156 |
| Chemical compound, drug | Acid-Phenol | Ambion | Am9722 |
| Commercial assay or kit | TRIzol | ThermoFisher | 15596026 |
| Recombinant DNA reagent | T4 Polynucleotide Kinase | New England Biolabs | M0201L |

*Continued on next page*

*Continued*

| Reagent type | Designation | Source or reference | Identifier |
|---|---|---|---|
| Chemical compound, drug | ATP, [γ−32P]- 3000 Ci/mmol 10mCi/ml EasyTide | PerkinElmer | BLU502A500UC |
| Peptide, recombinant protein | SuperScript IV Reverse Transcriptase | ThermoFisher | 18090010 |
| Peptide, recombinant protein | RNase H | New England Biolabs | M0297S |
| peptide, recombinant protein | TURBO DNase | ThermoFisher | AM2238 |
| Recombinant DNA reagent | KOD Hot Start DNA Polymerase | sigmaaldrich | 71086 |
| Commercial assay or kit | Micro Bio-Spin Columns With Bio-Gel P-6 in Tris Buffer | BioRad | 7326221 |
| Commercial assay or kit | Amylose Resin | New England Biolabs | E8021L |
| Commercial assay or kit | RNeasy Mini Kit | Qiagen | 74106 |
| Commercial assay or kit | SuperSignal West Femto Maximum Sensitivity Substrate | ThermoFisher | 34095 |
| Commercial assay or kit | Zero Blunt TOPO PCR Cloning Kit | invitrogen | K2800-20SC |
| Commercial assay or kit | FirstChoice RLM-RACE Kit | ThermoFisher | AM1700 |
| Other | Raw and analyzed RNA-seq data | This paper | GEO: GSE106168 https://www.ncbi.nlm.nih.gov/geo/query/acc.cgi?acc=GSE106168 |
| Other | Raw and analyzed LC-MS/MS data | This paper | PRIDE: PXD008128 |
| Other | Raw and analyzed RNA-seq data for GsrN-PP7hp purification | This paper | GEO: GSE106171 https://www.ncbi.nlm.nih.gov/geo/query/acc.cgi?acc=GSE106171 |
| Other | Raw and analyzed RNA-seq data for Network construction | (*Fang et al., 2013*) PMC3829707 | GEO: GSE46915 |
| Commercial assay or kit | Zeta-Probe Blotting Membranes | BioRad | 162–0165 |
| Commercial assay or kit | Low Molecular Weight Marker, 10–100 nt | Alfa Aesar | J76410 |
| Commercial assay or kit | Mini-PROTEAN TGX Precast Gel, 4–20% | BioRad | 456–1094 |
| Commercial assay or kit | Precision Plus Protein Kaleidoscope Prestained Protein Standards | BioRad | 1610375 |
| Sequence-based reagent | See *Supplementary file 1* | | |
| Sequence-based reagent | See *Supplementary file 1* | | |
| Software, algorithm | Bowtie2 | (*Langmead and Salzberg, 2012*) PMC3322381 | http://bowtie-bio.sourceforge.net/bowtie2/index.shtml |
| Software, algorithm | SAMTools | (*Li et al., 2009*) PMC2723002 | http://samtools.sourceforge.net/ |
| Software, algorithm | IntaRNA 2.0.2 | (*Mann et al., 2017*) 10.1093/nar/gkx279 | http://rna.informatik.uni-freiburg.de/IntaRNA/Input.jsp |
| Software, algorithm | Prism v6.04 | GraphPad Software, Inc. | https://www.graphpad.com/scientific-software/prism/ |
| Software, algorithm | WebLogo | (*Crooks et al., 2004*), PMC419797 | http://weblogo.berkeley.edu/logo.cgi |
| Software, algorithm | Geneious 11.0.2 | (*Kearse et al., 2012*), PMC3371832 | https://www.geneious.com/ |
| Software, algorithm | R v 3.3.3 | | https://www.r-project.org/ |
| Software, algorithm | Python v2.7 | | https://www.python.org/download/releases/2.7/ |
| Software, algorithm | Rockhopper 2.0 | (*Tjaden, 2015*), PMC4316799 | https://cs.wellesley.edu/~btjaden/Rockhopper/ |

*Continued on next page*

*Continued*

| Reagent type | Designation | Source or reference | Identifier |
|---|---|---|---|
| Software, algorithm | Edge-pro | (*Magoc et al., 2013*), PMC3603529 | http://ccb.jhu.edu/software/EDGE-pro/index.shtml |
| Software, algorithm | DESeq | (*Anders and Huber, 2010*), PMC3218662 | http://bioconductor.org/packages/release/bioc/html/DESeq.html |
| Software, algorithm | CLC Genomics Workbench 10 | (Qiagen) | https://www.qiagenbioinformatics.com/products/clc-genomics-workbench/ |
| Software, algorithm | MaxQuant | (*Cox et al., 2014*), PMC4159666 | http://www.coxdocs.org/doku.php?id=maxquant:start |
| Software, algorithm | IterativeRank | This paper | https://github.com/mtien/IterativeRank |
| Software, algorithm | Sliding_window_analysis | This paper | https://github.com/mtien/Sliding_window_analysis |

## Experimental model and subject details

### Growth media and conditions

*C. crescentus* was cultivated on peptone-yeast extract (PYE)-agar (0.2% peptone, 0.1% yeast extract, 1.5% agar, 1 mM $MgSO_4$, 0.5 mM $CaCl_2$) (*Ely, 1991*) supplemented with 1.5% agar at 30°C. Antibiotics were used at the following concentrations on this solid medium: kanamycin 25 µg/ml, tetracycline 2µg/ml, nalidixic acid 20 µg/ml, and chloramphenicol 2 µg/ml.

For liquid culture, *C. crescentus* was cultivated in either PYE or in M2X defined medium (*Ely, 1991*). PYE liquid: 0.2%(w/v) peptone, 0.1%(w/v) yeast extract, 1 mM $MgSO_4$, and 0.5 mM $CaCl_2$, autoclaved before use. M2X defined medium: 0.15% (w/v) xylose, 0.5 mM $CaCl_2$, 0.5 mM $MgSO_4$, 0.01 mM Fe Chelate, and 1x M2 salts, filtered with a 0.22 micron bottle top filter. One liter of 20x M2 stock was prepared by mixing 17.4 g $Na_2HPO_4$, 10.6 $KH_2PO_4$, and 10 g $NH_4Cl$. To induce gene expression from the *vanA* promoter, 500 µM vanillate (final concentration) was added. Antibiotics were used at the following concentrations in liquid medium: kanamycin 5 µg/ml, tetracycline 1 µg/ml, and chloramphenicol 2 µg/ml.

For cultivation of *E. coli* in liquid medium, we used lysogeny broth (LB). Antibiotics were used at the following concentrations: ampicillin 100 µg/ml, kanamycin 50 µg/ml, tetracycline 12 µg/ml, and chloramphenicol 20 µg/ml.

### Strain construction

All *C. crescentus* experiments were conducted using strain CB15 (*Poindexter, 1964*) and derivatives thereof. Plasmids were conjugated into CB15 (*Ely, 1991*) using the *E. coli* helper strain FC3 (*Finan et al., 1986*). Conjugations were performed by mixing the donor *E. coli* strain, FC3, and the CB15 recipient strain in a 1:1:5 ratio. Mixed cells were pelleted for 2 min at 15,000xg, resuspended in 100 µL, and spotted on a nonselective PYE-agar plate for 12–24 hr. Exconjugants containing the desired plasmid were selected on PYE agar containing the plasmid-specified antibiotic for selection and antibiotic nalidixic acid (20 µg/ml) to counter-select against both *E. coli* strains (helper and plasmid donor).

Gene deletion and nucleotide substitution strains were generated using the integrating plasmid pNPTS138 (*Ried and Collmer, 1987*). pNPTS138 transformation and integration occurs at a chromosomal site homologous to the insertion sequence in pNPTS138. Exconjugants with pNPTS138 plasmids were selected on PYE agar plates with 5 µg/ml kanamycin; 20 µg/ml nalidixic acid selected against the *E. coli* donor strain. Single colony exconjugants were inoculated into liquid PYE or M2X for 6–16 hr in a rolling 30°C incubator for non-selective growth. Nonselective liquid growth allows for a second recombination event to occur, which either restores the native locus or replaces the native locus with the insertion sequence that was engineered into pNPTS138. Counter-selection for the second recombination of pNPTS138 was carried out on PYE agar with 3% (w/v) sucrose. This selects for loss of the *sacB* gene during the second crossover event. Colonies were subjected to PCR genotyping and/or sequencing to identify to confirm the allele replacement.

Other strains utilized in this study originate from (*Herrou et al., 2010*), (*Purcell et al., 2007*), and (*Foreman et al., 2012*).

The Δ*gsrN* strains and Δ*sigT* strains were complemented by introducing the gene at an ectopic locus (either *vanA* or *xylX*) utilizing the integrating plasmids: pMT552, pMT674, and pMT680. pMT674 and pMT680 carry a chloramphenicol resistance marker gene (*cat*) and pMT552 carries a kanamycin resistance marker gene (*npt1*) (*Thanbichler et al., 2007*). pMT552 and pMT674 integrate into the *vanA* gene and pMT680 integrates into the *xylX* gene. Transformation of ectopic complementation plasmids conjugated (as described earlier). Introduction of *gsrN* complementation was done in the reverse direction of the inducible promoters. Introduction of *katG* was done in-frame in the same direction of the inducible promoters.

Replicating plasmids pPR9TT and pRKlac290 were conjugated as previously described earlier. pPR9TT and pRKlac290 were selected using tetracycline and chloramphenicol, respectively.

pMal-MBP-PP7CPHis was transformed into *E. coli* Rosetta by electroporation and plated on LB plates with ampicillin 100 µg/ml.

## Plasmid construction

Plasmid pNPTS138 was used to perform allele replacements and to generate gene deletions (*Ried and Collmer, 1987*; *West et al., 2002*). Primers for in-frame deletions and GeneBlocks (Gblocks) are listed in *Supplementary file 1*. Gene fragments were created by splice-overlap-extension and ligated into the digested pNPTS138 vector at restriction enzyme sites (HindIII, SpeI) or gene fragments were stitched together using Gibson assembly. pNPTS138 contains a *kan*[R] (*npt1*) antibiotic resistance marker and the counter-selectable marker gene *sacB*, which encodes levansucrase

Plasmids for *gsrN* genetic complementation experiments carried wild-type or mutant *gsrN* alleles cloned into the *vanA*locus where *gsrN* is antisense to the vanillate inducible v*anA* promoter. An in-frame stop codon was designed at a restriction enzyme site downstream of the *vanA* promoter to ensure translational read-through of the *vanA* transcript did not disrupt *gsrN* transcription. Tandem *gsrN* alleles (overexpression by multiple copies of *gsrN*) were constructed using Gblocks with unique ends for Gibson assembly into pMT552. Plasmids for genetic complementation of the *katG* mutant were constructed by cloning *katG* in-frame with the vanillate and xylose-inducible promoters of pMT674 and pMT680, respectively, at the NdeI and KpnI restriction sites. *katG* complementation plasmids did not include the 5' untranslated region (UTR) of *katG*.

Beta-galactosidase transcriptional and translational reporters utilized pRKlac290 (*Ely, 1991*) and pPR9TT (*Santos et al., 2001*) replicating plasmids, respectively. Transcriptional reporters of *gsrN* contained upstream and promoter sequences of *gsrN* cloned into the EcoRI and HindIII sites of pPRKlac290. Translational reporters of *katG* contained the 191 nucleotides 3' of the annotated *katG* transcriptional start site (*Zhou et al., 2015*) cloned into pPR9TT at HindIII and KpnI.

Protein expression plasmid pMal was used to express a maltose binding protein (MBP) fused to the N-terminus of a Pseudomonas Phage seven coat protein fused to a His-tag at its C-terminus (to generate MBP-PP7CP-His). The PP7CPHis protein sequence was amplified out of pET283xFlagPP7CPHis and inserted into pMal at SalI and EcoRI restriction sites. pET283xFlagPP7CPHis was a gift from Alex Ruthenburg and originates from Kathleen Collins (Addgene plasmid # 28174).

## Experimental method details

### Hydrogen peroxide/osmotic stress assays

Liquid cultures were passaged several times before stress treatment to insure that population growth rate and density was as consistent as possible prior to addition of hydrogen peroxide (oxidative stress) or sucrose (hyperosmotic stress). Briefly, starter cultures were inoculated in liquid M2X medium from colonies grown on PYE-agar plates. Cultures were grown overnight at 30°C in a rolling incubator. Overnight cultures were then diluted back to an optical density reading of 0.05 at 660 nm ($OD_{660}$ = 0.05) and grown in a rolling incubator at 30°C for 7–10 hr. After this period, cultures were re-diluted with M2X to $OD_{660}$ = 0.025 and grown overnight for 16 hr at 30°C in a rolling incubator. After this period, $OD_{660}$ was consistently 0.85–0.90. These cultures were then diluted to $OD_{660}$ = 0.05 and grown for 1 hr and split into two tubes. One tube received stress treatment and the other tube was untreated. Treated cultures were subjected to either hydrogen peroxide or sucrose.

For stress treatment, we used a freshly prepared 10 mM $H_2O_2$ solution diluted from a 30% (w/w) stock bottle (stock never more than 3 months old) or a stock of 80% (w/v) sucrose. The amount of 10 mM $H_2O_2$ added for stress perturbation depended on the volume of the culture and the desired final concentration of $H_2O_2$. Final volumes assessed in our studies are described for each experiment throughout this manuscript.

Treated cultures and untreated cultures were subsequently titered (10 µL sample in 90 µL of PYE) by initially diluting into 96-well plates. 5 µL spots from each dilution were plated on PYE-agar. Once spots dried, plates were incubated at 30°C for 2 days. Clearly visible colonies begin to form after 36 hr in the incubator.

The difference in colony forming units (CFU) between treated and untreated cultures was calculated using the following formula:

$$Relative\ CFU = \frac{Treated\ CFU \times 10^x}{Untreated\ CFU \times 10^y} \tag{1}$$

where x represents the countable (resolvable) dilution in which colonies are found in the treated sample dilution series and y represents the untreated sample dilution.

## β-galactosidase gene expression reporter assays

To assess reporter gene expression, liquid cultures were passaged several times as described in the hydrogen peroxide/osmotic stress assays section above. However, cultures were placed in a 30°C shaker instead of a 30°C rolling incubator. Exponential phase cultures were harvested when the last starter culture (i.e. the $OD_{660}$ = 0.05 culture made from the 16 hr overnight culture) reached an $OD_{660}$ of 0.2–0.25. Stationary growth cultures were harvested when the exponential phase culture reached an OD660 of 0.85–0.90. Reporter assays in which the effect of stress treatment was quantified were conducted on exponential phase cultures that were split immediately before treatment.

β-galactosidase activity from chloroform-permeabilized cells was measured using the colorimetric substrate o-nitrophenyl-β-D-galactopyranoside (ONPG). 1 mL enzymatic reactions contained 200–250 µL of chloroform-permeabilized cells, 550–600 µL of Z-buffer (60 mM $Na_2HPO_4$, 40 mM $NaH_2PO_4$, 10 mM KCl, 1 mM $MgSO_4$), and 200 µL of 4 mg/mL ONPG in 0.1 M $KPO_4$, pH 7.0. Chloroform-permeabilized cell samples were prepared from 100 to 150 µL of culture, 100 µL of PYE, and 50 µL of chloroform (chloroform volume is not included in the final calculation of the 1 mL reaction). Chloroform-treated cells were vortexed for 5–10 s to facilitate permeabilization. Z buffer and ONPG were added directly to chloroform-permeabilized cells. Reactions were incubated in the dark at room temperature and quenched with 1 mL of 1 M $Na_2CO_3$.

Each reporter construct was optimized with different reaction times and different volumes of cells. Reaction time and volume for each reporter was empirically determined by the development of the yellow pigment from chloroform-permeabilized *C. crescentus* CB15 cultures. Strains harboring the pRKlac290 transcriptional reporter plasmid containing the established GSR promoter reporter $P_{sigU}$ or $P_{gsrN}$ used 100 µL of cells and were quenched after 10 min and 18 min, respectively. Strains containing pRKlac290 with the *katG* promoter ($P_{katG}$) used 150 µL of cells and were quenched after 12 min. Strains with the translational reporter plasmid pPR9TT containing the 5'UTR of *katG* (wild-type and RS constructs) used 150 µL of cells and were quenched after 4 min.

Miller units were calculated as:

$$MU = \frac{A_{420} \times 1000}{A_{660} \times t \times v} \tag{2}$$

where $A_{420}$ is the absorbance of the quenched reaction measured at 420 nm on a Spectronic Genesys 20 spectrophotometer (ThermoFisher Scientific, Waltham, MA). $A_{660}$ is the optical density of the culture of cells used for the assay. t is time in minutes between the addition of ONPG to the time of quenching with $Na_2CO_3$. v is the volume in milliliters of the culture added to the reaction.

## TRIzol RNA extractions

Cultures used for the extraction of RNA were passaged in the same manner outlined in the hydrogen peroxide/osmotic stress assays section above. Exponential phase cultures were harvested from the last starter (i.e. the $OD_{660}$ = 0.05 culture at the 16 hr time point) when it reached an $OD_{660}$ of 0.20–

0.25. Stationary cultures were harvested when the final culture diluted to $OD_{660}$ = 0.025 reached an $OD_{660}$ of 0.85–0.90.

Exponential phase cultures ($OD_{660}$ of 0.20–0.25) harvested for extraction of RNA were pelleted at 15000xg for 3 min at ≈23°C (i.e. room temperature). Early stationary cultures ($OD_{660}$ of 0.85–0.90) were also pelleted at 15000xg for 30 s at ≈23°C. All media were aspirated using a vacuum flask. Cell pellets were resuspended in 1 mL of TRIzol. The TRIzol resuspension was heated for 10 min at 65°C, treated with 200 μL of chloroform and hand shaken. The chloroform mixture was allowed to stand for 5 min and then spun down at 15000xg for 15 min at 4°C. Approximately 500 μL of clear aqueous phase was extracted and mixed with 500 μL of 100% isopropanol. Samples were then incubated at −20°C overnight. Overnight isopropanol precipitation was then spun down at 15000xg for 15 min at 4°C. Isopropanol was aspirated, the pellet was washed in 1 mL of 75% ethanol, and sample was spun down at 15000xg for 15 min at 4°C. Ethanol was removed from pellet, and the pellet was left to dry for 15 min. The RNA pellet was resuspended in 25 μL of nuclease-free $H_2O$.

## Radiolabeled oligonucleotides

Oligonucleotides were radiolabeled with T4 Polynucleotide Kinase (PNK). 10 μL labeling reactions were composed of 1 μL of PNK, 1 μL PNK 10x Buffer, 2 μL of 5 μM oligonucleotides (1 μM final concentration), 4 μL $H_2O$, and 2 μL ATP, [γ−32P]. Reactions were incubated for a minimum of 37°C for 30 min. Total reactions were loaded onto a BioRad P-6 column to clean the reaction. Radiolabeled samples were stored at 4°C.

## Northern blots

RNA samples were resolved on a 10% acrylamide:bisacrylamide (29:1), 7 M urea, 89 mM Tris Borate pH 8.3, 2 mM $Na_2$EDTA (TBE) 17 by 15 cm gel, run for 1 hr and 50 min at 12 Watts constant power in TBE running buffer. The amount of sample loaded was between 1–5 μg of RNA, mixed in a 1:1 ratio with 2x RNA loading dye (9 M urea, 100 mM EDTA, 0.02% w/v xylene cyanol, 0.02% w/v bromophenol blue). Samples were heated for 8 min at 75°C and then subjected to an ice bath for 1 min before loading. Acrylamide gels with immobilized samples were then soaked in TBE buffer with ethidium bromide and imaged. Samples immobilized on the gel were transferred onto Zeta-Probe Blotting Membrane with a Trans-Blot SD Semi-Dry Transfer Cell. Transfer was done at 400 mA constant current with voltage not exceeding 25V for 2 hr. Membrane was then subjected to two doses of 120 mJ/cm$^2$ UV radiation, using a Stratalinker UV cross-linker. Membranes were subsequently prehybridized 2 times for 30 min in hybridization buffer at 65°C in a rotating hybridization oven. Hybridization buffer is a variation of the Church and Gilbert hybridization buffer (20 mM sodium phosphate, pH 7, 300 mM NaCl, 1% SDS). Blots were hybridized with hybridization buffer containing the radiolabeled oligonucleotide probes described above. Hybridization buffer was always prepared so that GsrN probe concentration was approximately 1 nM, 5S rRNA probe concentration was approximately 2 pM, and tRNA-Tyr probe was 500 pM. Hybridization took place over 16 hr at 65°C in a rotating hybridization oven. Membranes were then incubated with wash buffer three times for 20 min at 65°C in a rotating hybridization oven. Wash buffer contained 20 mM sodium phosphate (pH 7.2), 300 mM NaCl, 2 mM EDTA, and 0.1% SDS. Membranes were then wrapped in plastic wrap and placed directly against a Molecular Dynamics Phosphor Screen. Screens were imaged with Personal Molecular Imager (PMI) System. Membrane exposure time was determined using a Geiger counter: 100 × 2 min, 10 × 30–60 min, 1.0 × 8–16 hr, 0.1 × 48–72 hr.

Intensity of GsrN bands or *katG* mRNA dots was calculated by dividing the probe signal specific to GsrN or *katG* mRNA over the probe signal specific to the 5S rRNA multiplied by 100. Normalization of *katG* mRNA specific probes in the dot blot was carried out in a manner similar to that described for Northern blot, in which the 5S rRNA probe signal was used for normalization.

$$Normalized\ volume_x = \frac{volume_x(CNT * mm^2)}{volume_{5s\ rRNA\ probe}(CNT * mm^2)} \quad (3)$$

## Rifampicin transcription inhibition assays

Liquid *C. crescentus* CB15 cultures were passaged in the same manner outlined in the hydrogen peroxide/osmotic stress assays section. However, cells for transcription inhibition assays were grown to an $OD_{660}$ of 0.2–0.25 from the last starter culture (i.e. inoculated from the $OD_{660}$ = 0.05 culture from

16 hr growth) and split across six tubes and labeled: untreated, 30 s treatment, 2 min treatment, 4 min treatment, 8 min treatment, and 16 min treatment. Untreated cultures were the 0 time point where no rifampicin was added. Rifampicin treated cultures were subjected to a final concentration of 10 µg/mL (from a 10 mg/mL stock in methanol) and were grown in a rolling incubator at 30°C. The 30 s rifampicin treatment refers to the centrifugation time (15000xg for 30 s at room temperature) to pellet the cells. Thus, the 30 s sample was immediately pelleted after exposure to rifampicin. 2 min, 4 min, 8 min, and 16 min samples were placed into a rolling incubator after exposure and were removed 30 s prior to their indicated time point, (i.e. 2 min culture was removed from the incubator at 1 min and 30 s). Pellets were then subjected to TRIzol extraction as described earlier. RNA extracts were subjected to Northern Blot analysis as described earlier.

Intensity of full-length and 5'isoform of GsrN bands were first adjusted to the intensity of the 5S rRNA control, as described in *Equation 3*. To plot the GsrN decay curve, all adjusted bands were then divided by the intensity of the 0 time point (untreated culture) and plotted in Prism v6.04.

$$Normalized\ timepoint\ volume_t = \frac{Normalized\ volume_t}{Normalized\ volume_0} \qquad (4)$$

## Primer extension

Primer extension was carried out using the SuperScript IV Reverse Transcriptase standalone enzyme. Total RNA from $gsrN^{++}$ and $\Delta gsrN$ strains was extracted from stationary cultures ($OD_{660}$ = 0.95–1.0) as described in the TRIzol extraction section. Primers for extension were first HPLC purified (Integrated DNA technologies) and radiolabeled as described in the Radiolabeled Oligonucleotides section.

Briefly, 14 µL annealing reactions comprised of the following final concentrations/amounts: 0.1 µM of gene specific radiolabeled primer, 0.3–0.5 mM of dNTPs, 2 µg of total RNA, and when necessary 0.5 mM ddNTPs. ddNTP reactions had a 3 dNTP:5 ddNTP ratio and were conducted using total RNA from $gsrN^{++}$. Annealing reactions were incubated at 65°C for 5 min and subsequently incubated on ice for at least 1 min.

Extension reactions contained 14 µL annealing reactions with 6 µL of SuperScript IV Reverse Transcriptase master mix (final concentrations/amount 5 mM DTT, 2.0 U/µL, 1x SSIV buffer). Reactions were incubated at 50–55°C for 10 min and then incubated at 80°C for 10 min to inactivate the reaction.

After the extension reaction, 1 µL of RNase H was added to the mixture. This was incubated at 37°C for 20 min and mixed with 20 µL of 2x RNA loading dye. Reactions were subsequently heated for 8 min at 80°C, subjected to an ice bath for 1 min, and loaded onto a 33.8 by 19.7 cm 20% acrylamide:bisacrylamide gel (as outlined in the Northern Blot section). Reactions were loaded on the gel along with a labeled Low Molecular Weight Marker (10–100 nt; Affymetrix/USB). Final amounts loaded were estimated using a Geiger counter, such that 10 mR/hr was loaded for each sample. Primer extension samples were resolved on the gel at 10 Watts constant power until unextended primer reached the bottom of the gel. The acrylamide gel was wrapped in plastic, exposed, and imaged as outlined in the Northern Blot section.

## 5' RACE of GsrN

Rapid amplification of cDNA 5'ends of GsrN was carried out using components of the FirstChoice RLM-RACE Kit and the SuperScript IV Reverse Transcriptase standalone enzyme. Cloning of cDNA library was carried out with the Zero Blunt TOPO PCR Cloning Kit. Total RNA from $gsrN^{++}$ strains was extracted from stationary phase cultures ($OD_{660}$ = 0.95–1.0) as described in the TRIzol extraction section.

Briefly, 10 µL Tobacco Acid Pyrophosphatase (TAP) reactions used 5 µg of total RNA with 2 µL of TAP and 1 µL of TAP buffer with remaining volume comprised of Nuclease-free water. Reactions were incubated at 37°C for 1 hr. TAP-treated samples were then subjected to ligation in parallel with no-TAP total RNA samples. Tap RNA sample ligation reactions (10 µL) follow: 2 µL of TAP treated RNA, 1 µL of 5'RACE adaptor, 1 µL of T4 RNA Ligase, 1 µL 10X T4 RNA Ligase Buffer, and 4 µL Nuclease-free water. No-TAP RNA sample ligation reactions (10 µL) follow: 3 µg of untreated total RNA, 1 µL of 5'RACE adaptor, 1 µL of T4 RNA Ligase, 1 µL 10X T4 RNA Ligase Buffer, and remaining volume of Nuclease-free water. Reactions were incubated at 37°C for 1 hr.

For the reverse transcription reaction (first strand synthesis), we used the SuperScript IV reverse transcriptase and the target-specific primer (NB12, see *Supplementary file 1*). The 20 μL reaction follows: 4 μL of ligated RNA, 4 μL of dNTP, 2 μL of 1 μM primer, 4 μL RT-Buffer, 1 μL of SSIV-RT, 3 μL water, and 2 μL of fresh 100 mM DTT. Reactions were incubated at 55°C for 10 min then 80°C for 10 min (to deactivate).

For second strand synthesis and amplification, we used KOD Hot Start DNA Polymerase with the 5'RACE inner primer complementary to the adapter and a nested target-specific primer (1189 in *Supplementary file 1*). The 25 μL reactions follow: 12.5 μL 2X Buffer, 0.5 μL KOD Polymerase, 5 μL of 2 mM dNTP, 2.5 μL of 50% DMSO, 1.5 μL of 5 mM forward primer, 1.5 μL of 5 mM reverse primer, and 1.5 μL of reverse transcribed 1$^{st}$ strand synthesis cDNA. Reaction protocol follows: 3 min 95°C incubation, followed by a 35 cycled reaction consisting of a 15 s 95°C melting step, a 15 s 60°C annealing step, and a 30 s 68°C extension step, and lastly a 1 min 68°C extension step.

PCR products were blunt-cloned using the Zero Blunt TOPO PCR Cloning Kit. First a 5 μL pre-reaction mix consisting of 2 μL PCR product, 1 μL kit salt solution, and 2 μL water was prepared. 1 μL of the pCR-Blunt II-TOPO was then added to the pre-reaction mix and incubated at room temperature for 5 min and then immediately put on ice. Ligation reaction was then incubated with ice-thawed chemically competent *E. coli* cells for 5 min. Cells were heat shocked for 30 s at 42°C, then incubated on ice for 5 min. 250 μL of SOC media was then added to the cells and incubated 37°C in a shaking incubator. 50 microliters of outgrown cells were placed on LB-Kanamycin plates with an antibiotic concentration of 50 μg/mL. Single colonies were grown overnight and sequenced with M13F and M13R primers provided in the Zero Blunt TOPO PCR Cloning Kit.

Sequences were submitted to the University of Chicago Comprehensive Cancer Center DNA Sequencing and Genotyping Facility. Chromatograph traces were analyzed with Geneious 11.0.2. Traces were subjected to mapping and trimming of the 5'RACE inner primer/adaptor sequence and the flanking regions used for blunt-cloning. Trimmed sequences are presented in *Figure 3—figure supplement 1E*.

## Affinity purification of GsrN using a PP7hp-PP7cp system

GsrN constructs containing a Pseudomonas phage 7 RNA hairpin (PP7hp) sequence were affinity purified using a hairpin-binding phage coat protein (PP7cp) immobilized on agarose beads. To prepare the coat protein, a 50 mL culture of *E. coli* Rosetta carrying an expression plasmid for PP7cp fused to maltose binding protein (MBP) at its N-terminus and a His-tag at its C-terminus (pMal-PP7cp-HIS) was grown at 37°C in a shaking incubator overnight in LB-ampicillin broth. Overnight cultures were rediluted and grown to OD$_{600}$ = 0.6. Cells were then induced with 1 mM IPTG for 5 hr and spun down at 8000 g at 4°C for 10 min. The cell pellet was resuspended in 6 mL of ice-cold lysis buffer (125 mM NaCl, 25 mM Tris-HCl pH 7.5, 10 mM Imidazole) and mechanically lysed in a LV1 Microfluidizer. Lysate was immediately added to 500 μL of amylose resin slurry that was prewashed with ice-cold lysis buffer. After the sample was loaded, beads were washed in 50x bead volume (~10 mL) of ice-cold lysis buffer.

A 50 mL culture of *C. crescentus* Δ*gsrN* carrying plasmid pMT552 expressing PP7hp-tagged alleles of *gsrN* was grown at 30°C in a shaking incubator overnight in M2X medium. The culture was prepared from a starter and passaged as outlined in the hydrogen peroxide/osmotic stress assays section. Cells were grown to an OD$_{660}$ = 0.85–0.90. Cells were spun down at 8000 g at 4°C for 15 min, resuspended in 6 mL of ice-cold lysis buffer, and mechanically lysed in a LV1 Microfluidizer. Lysate was immediately loaded onto a column of amylose resin on which MBP-PP7cp-HIS had been immobilized. After the sample was loaded, beads were washed in 50x bead volume (~10 mL) of ice-cold lysis buffer. Elution of MBP-PP7cp-HIS bound to GsrN-PP7hp and associated biomolecules was completed over three 0.5 mL elution steps using 500 mM maltose. Each 0.5 mL elution was then mixed with equal volumes of acid-phenol for RNA extraction for RNA analysis, or equal volumes of SDS-Loading Buffer (200 mM Tris-HCl pH 6.8, 400 mM DTT, 8% SDS, 0.4% bromophenol blue, 40% glycerol) for protein analysis. For the RNA analysis, the three elution fractions were combined in an isopropanol precipitation step. RNA samples were subjected to DNase treatment as outlined in the RNA-seq sequencing section.

## Acid-phenol RNA extraction

Samples for acid-phenol extractions were mixed with equal volumes of acid-phenol and vortexed intermittently at room temperature for 10 min. Phenol mixture was spun down for 15 min at maximum speed at 4°C. The aqueous phase was extracted, cleaned with an equal volume of chloroform, and spun down for 15 min at maximum speed at 4°C. The aqueous phase was extracted from the organic and equal volumes of 100% isopropanol were added. Linear acrylamide was added to the isopropanol precipitation to improve pelleting (1 μL per 100 μL of isopropanol sample). Samples were then incubated at −20°C overnight and spun down at 15,000xg for 15 min at 4°C. The isopropanol was aspirated, the pellet washed in 1 mL of 75% ethanol, and sample spun again at 15,000xg for 15 min at 4°C. Ethanol was removed from the RNA pellet, and pellet was left to dry for 15 min. Pellet was resuspended in 25 μL of nuclease-free $H_2O$.

## RNA dot blot analysis

Samples ($\approx$ 3 μg) for dot blot analysis were mixed with equal volumes of 2x RNA loading dye as in a Northern blot, and heated for 8 min at 75°C. Samples were then spotted on a Zeta-Probe Blotting Membrane and left to dry for 30 min. Spotted membrane was then subjected to two doses of 120 mJ/cm$^2$ UV radiation (Stratalinker UV crosslinker). The membrane was then prehybridized two times for 30 min in hybridization buffer at 65°C in a rotating hybridization oven. After pre-hybridization, we added radiolabeled oligonucleotide probes. Hybridization buffer with probes was always prepared so that each probe's concentration was approximately 1 nM. *katG* mRNA was first hybridized for 16 hr at 65°C in a rotating hybridization oven. Membrane was then washed with wash buffer three times, 20 min each at 65°C in a rotating hybridization oven. The blot was exposed for 48 hr to a Molecular Dynamics Phosphor screen and imaged on a Personal Molecular Imager as described above. Membrane was subsequently stripped with two rounds of boiling in 0.1% SDS solution and incubated for 30 min at 65°C in a rotating hybridization oven. Following stripping, the membrane was subjected to two rounds of prehybridization and then hybridized for 16 hr at 65°C in a rotating hybridization oven with the probe specific to the 5' end of GsrN. Membrane was then washed again with wash buffer three times for 20 min each at 65°C in a rotating hybridization oven. This GsrN blot was exposed for 36 hr to the phosphor screen and imaged. The membrane was stripped four times after GsrN probe exposure. Following stripping, membrane was again subjected to two rounds of prehybridization and then hybridized for 16 hr at 65°C in a rotating hybridization oven with the probe specific to 5S rRNA. Membrane washed with Wash Buffer three times, 20 min each at 65°C in a rotating hybridization oven. This 5S RNA blot was exposed to the phosphor screen for 1 hr and imaged.

## Western blot analysis

Strains from which protein samples were prepared for Western blot analysis were grown and passaged as outlined in the hydrogen peroxide/osmotic stress assays section. However, cultures were taken from the overnight 16 hr growth when OD$_{660}$ reached 0.85–0.90. 1 mL of these cultures was then pelleted, resuspended in 125 μL of Western blot buffer (10 mM Tris pH 7.4, 1 mM CaCl$_2$, and 5 μg/mL of DNase), and mixed with 125 μL SDS-Loading buffer. Samples were boiled at 85°C for 10 min, and 10–20 μL of each sample was loaded onto a Mini-PROTEAN TGX Precast Gradient Gel (4–20%) with Precision Plus Protein Kaleidoscope Prestained Protein Standards. Samples were resolved at 35 mA constant current in SDS running buffer (0.3% Tris, 18.8% Glycine, 0.1% SDS). Gels were run until the 25 kDa marker reached the bottom of the gel. Gel was transferred to an Immobilon-P PVDF Membrane using a Mini Trans-Blot Cell after preincubation in Western transfer buffer (0.3% Tris, 18.8% Glycine, 20% methanol). Transfer was carried out at 4°C, 100 V for 1 hr and 20 min in Western transfer buffer. The membrane was then blocked in 5% (w/v) powdered milk in Tris-buffered Saline Tween (TBST: 137 mM NaCl, 2.3 mM KCl, 20 mM Tris pH 7.4, 0.1% (v/v) of Tween 20) overnight at room temperature on a rotating platform. Primary incubation with a DYKDDDDK(i.e. M2)-Tag Monoclonal Antibody (clone FG4R) was carried out for 3 hr in 5% powdered milk TBST at room temperature on a rotating platform (4 μL antibody in 12 mL). Membrane was then washed three times in TBST for 15 min each at room temperature on a rotating platform. Secondary incubation with Goat anti-Mouse IgG (H + L) Secondary Antibody, HRP was for 1 hr at room temperature on a rotating platform (3 μL antibody in 15 mL). Finally, membrane was washed three times in TBST for 15 min

each at room temperature on a rotating platform. Chemiluminescence was performed using the SuperSignal West Femto Maximum Sensitivity Substrate and was imaged using a ChemiDoc MP Imaging System version 6.0. Chemiluminescence was measured using the ChemSens program with an exposure time of ~2 min.

Western blot lane normalization of KatG-M2 specific bands was conducted by normalizing total signal from the doublet signal in the M2 specific background to that of the non-specific band (found in strains were there was no M2 tagged KatG). Samples extracted on the same day were run on the same gel. Lane normalized samples were then normalized to the levels of KatG-M2 signal in the wild-type untreated samples for that specific gel.

$$Lane\ Normalized\ volume_x = \frac{Volume\ of\ top_x + Volume\ of\ bottom_x}{Volume\ of\ non-specific_x} \tag{5}$$

$$WT\ Normalized\ volume_x = \frac{Lane\ Normalized\ volume_x}{Lane\ Normalized\ volume_{untreated\ WT}} \tag{6}$$

## RNA-seq preparation

Total RNA was extracted from cultures passaged similarly to the hydrogen peroxide/osmotic stress assays section. However, cultures were harvested at $OD_{660}$ = 0.85–0.90 from the 16 hr overnight growth. Total RNA extraction followed the procedure outlined in the TRIzol extraction section. Resuspended RNA pellets after the 75% ethanol wash were loaded onto an RNeasy Mini Kit column (100 μL sample, 350 μL RLT, 250 μL 100% ethanol). Immobilized RNA was then subjected to an on-column DNase digestion with TURBO DNase. DNase treatment was repeated twice on the same column; each incubation was 30 min at 30°C with 70 μL solutions of DNase Turbo (7 μL DNase, 7 μL 10x Buffer, 56 μL diH$_2$O). RNA was eluted from column, rRNA was depleted using Ribo-Zero rRNA Removal (Gram-negative bacteria) Kit (Epicentre). RNA-seq libraries were prepared with an Illumina TruSeq stranded RNA kit according to manufacturer's instructions. The libraries were sequenced on an Illumina HiSeq 4000 at the University of Chicago Functional Genomics Facility.

## Soluble protein extraction for LC-MS/MS proteomics

Total soluble protein for proteomic measurements was extracted from cultures passaged similarly to the hydrogen peroxide/osmotic stress assays section. However, harvested cultures were grown to an $OD_{660}$ = 0.85–0.90 in 50 mL of M2X during the 16 hr overnight growth in a 30°C shaking incubator. Cells were spun down at 8000 g at 4°C for 15 min. Cells were resuspended in 6 mL of ice-cold lysis buffer. Cells were mechanically lysed in LV1 Microfluidizer. Lysate was then spun down at 8000 g at 4°C for 15 min. Protein samples were resolved on a 12% MOPS buffered 1D Gel (Thermo Scientific) for 10 min at 200V constant. Gel was stained with Imperial Protein stain (Thermo Scientific), and a ~ 2 cm plug was digested with trypsin. Detailed trypsin digestion and peptide extraction by the facility is published in *Truman et al. (2012)*.

## LC-MS/MS data collection and analysis

Samples for analysis were run on an electrospray tandem mass spectrometer (Thermo Q-Exactive Orbitrap), using a 70,000 RP survey scan in profile mode, m/z 360–2000 Fa, with lockmasses, followed by 20 MSMS HCD fragmentation scans at 17,500 resolution on doubly and triply charged precursors. Single charged ions were excluded, and ions selected for MS/MS were placed on an exclusion list for 60 s (*Truman et al., 2012*).

## Computational methods

### Network construction

RNAseq data (15 read files) was obtained from the NCBI GEO database from (*Fang et al., 2013*). Read files are comprised of 3 biological replicates of total RNA extracted from *C. crescentus* cultures at five time points across the cell cycle (0, 30, 60, 90, and 120 min post synchrony). Reads were mapped and quantified with Rockhopper 2.0 (*Tjaden, 2015*). The estimated expression levels of each gene across the five time points were extracted from the 'Expression' column in the '_transcripts.txt' file, using the 'verbose' output. Expression of each gene across the five time points was

normalized using python scripts as follows: for a given gene, the normalized expression of the gene at a time point, t, is divided by the sum of the gene's expression across all the time points, *Equation 7*. Thus, the sum of a gene's normalized expression across the five time points would equal 1.

$$Let \; t \in T, \; where \; T = \{0, 30, 60, 90, 120\}$$

$$Normalized \, Transcript_t = \frac{Expression_t}{\sum^T Expression_t} \tag{7}$$

We computed Pearson's correlation coefficient based on normalized expression between all pairwise combinations of genes. Correlation coefficients were organized into a numpy.matrix data structure where each row and column corresponds to the same gene order. Correlation coefficients less than 0 were not considered for this analysis and were assigned the value 0. We refer to this matrix as the Rho-matrix. The Rho-matrix is symmetric and the product of its diagonal is 1. The Rho-matrix represents the weighted edges of the network, where the value of 0 demonstrates no edge is drawn between nodes.

A one-dimensional weight matrix that corresponds to the rows and columns of the Rho-matrix was constructed as a numpy.matrix data structure with all values initialized at 0. Lastly, a key array was constructed in conjunction with the Rho-matrix and weight-matrix for initializing the assignment of weight and obtaining the final weights of the algorithm. The weight-matrix represents the weight of the nodes of the network and the key matrix represents the gene name of the node.

## Iterative ranking: matrices and algorithms

Iterative ranking algorithms are a class of analytical tools used to understand relationships between nodes of a given network. The iterative ranking algorithm used to dissect the general stress response in the transcription-based network follows:

$$f^t = \propto f^0 + (1 - \propto) P f^{t-1} \tag{8}$$

Given the Rho-matrix (P) and weight-matrix ($f$), the weight-matrix after t-iterations is *Equation 8*.

For *Equation 8*, let $\propto$ represent a dampening factor applied to the initialize $(t) = 0$ weight of the nodes, $f^0$. The final weights of the weight-matrix as $t \to \infty$ converge to a stable solution, *Equation 9*.

$$f^\propto = \propto [1 - (1 - \propto)P]^{-1} f^0 \tag{9}$$

Algorithm and solution information was adapted from (*Wang and Marcotte, 2010*).

Initial weight-matrix, ($f^0$), was created by assigning the weight 1.0 to the corresponding positions of the seven genes known to regulate the General Stress Response (GSR) of *C. crescentus*: *sigT*, *phyR*, *phyK*, *sigU*, *nepR*, *lovR*, and *lovK*. Normalization of the values of the Rho-matrix, P, was performed by normalizing each column such that each column has a sum equal to one and then repeating the same normalization process by rows.

## Iterative rank parameter tuning

Iterative rank parameters were optimized through the self-prediction of known associated components of the General Stress Response (GSR). Variables tuned for exploration were the $\propto$ parameter and the reduction of the number of edges based on correlation cut-offs. We chose to base our parameters on which condition best predicted the gene *phyR*, when initializing the weight-matrix with *sigT*, *sigU*, *nepR*, *phyK*, *lovR*, and *lovK* values of 1. Varying these two parameters showed that an edge reduction of ρ >0.9 and an alpha factor greater than 0.5 yielded the highest rank for *phyR* (*Figure 1—figure supplement 1A*).

A ρ >0.9 edge reduction reduces the number of edges each node has (*Figure 1—figure supplement 1B*). The total number of edges was reduced from 10225998 edges to 946558 (*Figure 1—figure supplement 1C*). Only 19 nodes (.46%) were completely disconnected from the network (zero number of edges). Tuning script is available at https://github.com/mtien/IterativeRank (*Tien, 2017a*).

## Identification of σ$^T$-promoter motifs

Motif finder utilized a python script that scans 200 nucleotides upstream of annotated transcriptional start sites (*Zhou et al., 2015*) or predicted translational start sites (TSS) (*Marks et al., 2010*).

We built a simple python library to take in genomic FASTA files, find specified regions of interest, and extract 200 nucleotides from a given strand. We used the *Caulobacter crescentus* NA1000 annotation (CP001340) from NCBI as the input genomic file and used the predicted TSS (when available) or annotated gene start sites as the region and strand specifier. After locating the position and strand within the file, we extracted the 200 nucleotides directly upstream of the site of interest and put the regions into a character-match calculator. Our simple calculator reported a list of positions for −35 (GGAAC) and for −10 elements (CGTT) of σ$^T$-dependent promoters within the 200-nucleotide input string. Only strict matches to these elements were reported. Spacers were calculated between all pairwise −35 and −10 matches. We identified potential σ$^T$-dependent promoters by identifying consensus −35 to −10 sequences with 15–17 base spacing. Sequence logos were generated from (*Crooks et al., 2004*)

## IntaRNA analysis

IntaRNA version 2.0.2 is a program within the Freiberg RNA Tools collection (*Mann et al., 2017*). To predict likely RNA-RNA associations between predicted unstructured regions within GsrN and its RNA targets, we input the sequence of GsrN as the query ncRNA sequence and a FASTA file of either: 1) windows significantly enriched in the GsrN(37)-PP7hp purification from our sliding window analysis with an additional 100 base pairs (50 bp on each side of the window) or 2) entire gene windows that showed significant enrichment from our Rockhopper analysis (*Figure 5—source data 3*).

Output from IntaRNA 2.0.2 comprised a csv file of target binding sites and the corresponding GsrN binding sites. We extract the predicted binding sites of the targets with a python script and parsed the targets into those predicted to bind the first exposed loop and the second exposed loop. Sequence logos were generated from (*Crooks et al., 2004*)

## Phylogenetic tree construction

A 16S rRNA phylogenetic tree of Alphaproteobacteria was constructed by extracting 16S rRNA sequences for all species listed in *Figure 9* and using the tree building package in Geneious 11.0.2 (*Kearse et al., 2012*). The tree was constructed using a global alignment with free end gaps and a cost matrix of 65% similarity (5.0/~4.0). The genetic distance model was the Tamura-Nei and the tree building method employed was neighbor-joining. *E. litoralis* was the out-group for tree construction.

## Prediction of *gsrN* homologs

A homology search based on the sequence of GsrN was conducted using BLASTn (*Altschul et al., 1990*). This simple search provided a list of clear GsrN homologs in the Caulobacteraceae family (*Caulobacter*, *Brevundimonas*, and *Phenylobacterium*).

Identification of homologs in other genera relied on analysis of published transcriptomic data, searching specifically for gene expression from intergenic regions. Analyzed data included *Rhizobium etli* (*Jans et al., 2013*), *Sinorhizobium meliloti* (*Valverde et al., 2008*) and *Brucella abortus* (*Kim et al., 2014*). The prediction of GsrN homologs in *Rhodopseudomonas palustris* and *Bradyrhizobium diazoefficiens* is completely based on the proximity of a GsrN-like sequence to the GSR locus and the presence of a σ$^{ecfG}$-binding site in the predicted promoters of these predicted genes.

## Mapping reads from RNA-seq data

RNA-seq read files (fastQ) were aligned with sequence files (fastA) using bowtie 2.0 (*Langmead and Salzberg, 2012*). SAMTools was then used to calculate the depth and coverage of each nucleotide in the hit output file from bowtie 2.0 (*Li et al., 2009*). Normalization of reads per nucleotide was computed by normalizing each count to the per million total number of reads mapped to all of the CP001340.1 genome. Normalized reads per nucleotide was then plotted in Prism v6.04 where standard error and mean were calculated.

## RNA-seq analysis of mRNAs that co-elute with GsrN

RNA-seq read files (fastQ) from the three replicate GsrN(37)::PP7hp purifications and duplicate PP7hp-GsrN-3' purifications were quantified and analyzed with Rockhopper 2.0 (*Tjaden, 2015*). Reads were mapped to modified *C. crescentus* genome files (fastA, PTT, RNT) where the wild-type *gsrN* locus was replaced with the sequence of *gsrN(37)-PP7hp*. Using the 'verbose output' option and the resulting 'transcripts.txt' file, we pruned the dataset to find genes that had low FDR values ('qValue'<0.05), were significantly enriched in GsrN(37)::PP7 ('Expression GsrN(37)-PP7hp' > 'Expression PP7hp-GsrN-3''), and had a high total number of reads that mapped to GsrN(37)::PP7 ('Expression GsrN(37)-PP7hp'>1000). This analysis provided a list of 35 candidate genes (*Figure 5—source data 1*).

The Rockhopper analysis package organizes reads into IGV (integrative Genomic Viewer) files. Upon visual inspection and spot validation of the 35 candidates in IGV, we found 26 genes with consistently higher signal across the three GsrN(37)::PP7hp purifications relative to PP7hp-GsrN-3' control fractions. In some cases, reads mapped outside coding sequences. Such reads mapped proximal to the 5' end of annotated genes and to intergenic regions. We observed uneven read distribution across some annotated genes. Cases in which reads were not evenly distributed across a gene were typically not classified as significantly different from the control samples in 'Expression' or 'qValue' by Rockhopper even when a clear bias in read density was visually evident (most often at the 5' end of the gene).

As a second approach, we performed a systematic window annotation analysis to capture the unaccounted read density differences between the two purified fractions (GsrN(37)::PP7hp and the PP7hp-GsrN-3' negative control). Windows were generated by in silico fragmentation of the *C. crescentus* NA1000 genome sequence, designating 25 base pair windows across the genome. We prepared new annotated window files (FASTA, PTT, RNT) for wild-type, *gsrN(37)-PP7hp*, and *PP7hp-gsrN-3'*. The window identification number corresponds to the same sequence across the three different FASTA sequences.

Mapping and quantification of reads to these windows was conducted using the EDGE-pro analysis pipeline (*Magoc et al., 2013*). A caveat of EDGE-pro quantification is the potential misattribution of reads to input windows. EDGE-pro quantification does not take strand information into account when mapping reads to input windows.

Read quantification of the *gsrN(37)::PP7hp* purifications showed consistent differences in one of the three samples. *gsrN(37)::PP7hp* sample 1 contained 2.69% reads mapped to *gsrN(37)-PP7hp* while samples 2 and 3 had 15.78% and 14.04% mapped to *gsrN(37)-PP7hp* respectively. Additionally, we observed that sample one had several genes that were strongly enriched in sample one and not in sample 2 and 3. Thus we employed a metric to balance the discrepancies between the three separate purifications. To minimize potential false positives, we calculated the average of all three samples and the average of samples 2 and 3. If the total average was 1.5 times greater than the samples 2 and 3 average, we assumed that the sample one artificially raised the average RPKM value and did not consider any data from any of the purifications in that specific window. The total window population decreased from 161713 windows to 109648 windows after this correction. This process is reflected in the https://github.com/mtien/Sliding_window_analysis script 'remove_high_variant_windows.py' (*Tien, 2017b*).

From the RPKM values calculated with EDGE-pro, we used the R-package, DESeq (*Anders and Huber, 2010*), to assess statistically significant differences between windows of expression. Candidate windows enriched in the GsrN(37)::PP7 fractions were identified using metrics similar to what is applied to traditional RNA-seq data. Briefly, we identified windows that had a low p-values (pvalue <0.10), were enriched in the GsrN(37)::PP7 ('baseMean GsrN(37)-PP7hp' > 'baseMean PP7hp'), and had a high level of reads mapped to the gene in the GsrN(37)::PP7 ('baseMean GsrN (37)-PP7hp'>1000) (*Figure 5—source data 2*). Since the read density of windows from the total RNA extracted from the PP7-purification did not converge when estimating dispersion with a general linear model, we added total RNA seq read density from wild-type strains grown in stationary phase to help model the dispersion for the negative binomial analysis by DESeq, GSE106168.

Adjacent significant windows were then combined and mapped onto the annotated genome of *C. crescentus*. In order to correct for strand information lost in EDGE-pro quantitation, bowtie file

information was used to define the strand of reads mapped to combined significant windows (*Table 1*).

## RNA-seq processing of total RNA

Analysis of whole genome RNA-seq data was conducted using the CLC Genomics Workbench (Qiagen). Reads were mapped to the *C. crescentus* NA1000 genome (accession CP001340.1) (*Marks et al., 2010*). Differential expression was determined using Wald test in the CLC Workbench suite (*Figure 8—source data 2*).

## LC-MS/MS processing of total soluble protein

Raw files of LC-MS/MS data collected on wild-type, Δ*gsrN, and gsrN*++ were processed using the MaxQuant software suitev1.5.1.2 (*Cox et al., 2014*). Samples were run against a FASTA file of proteins from the UniProt database (UP000001364) and standard contaminants. The label free quantitation (LFQ) option was turned on. Fixed modification included carbamidomethyl (C) and variable modifications were acetyl or formyl (N-term) and oxidation (M). Protein group files were created for three comparisons: wild-type versus Δ*gsrN*, Δ*gsrN* versus *gsrN*++, and wild-type versus *gsrN*++ samples.

LFQ values for each protein group were compiled across all three runs and used as estimated protein quantities in our analyses (*Figure 8A*). Each strain had a total of 6 LFQ values for every protein group, two from each of the comparisons. Average LFQ values were only calculated if three or more LFQ values were found for a given protein group. This allowed for protein groups that had a sufficient amount of signal across all the samples and analyses to be considered for comparison. Once averages for each protein group were calculated, we calculated the fold change between samples from different backgrounds by dividing the averages and taking the log-2 transformation (log2Fold).

Multiple t-tests were conducted using all 6 LFQ value obtained across the three MaxQuant runs. We used the multiple t-test analysis from GraphPad Prism version 6.04 for MacOS, GraphPad Software, La Jolla California USA, www.graphpad.com. The false discovery rate (Q) value was set to 5.000% and each row was analyzed individually, without assuming a consistent SD.

## Data and software availability

IterativeRank and RhoNetwork python libraries are available on https://github.com/mtien/IterativeRank. A copy is archived at https://github.com/elifesciences-publications/IterativeRank.

Scripts used to analyze the total RNA reads from the PP7-affinity purification are available on https://github.com/mtien/Sliding_window_analysis. A copy is archived at https://github.com/elifesciences-publications/Sliding_window_analysis.

RNA-seq data of wild-type, Δ*gsrN*, and *gsrN*++ early early stationary cultures are deposited in the NCBI GEO database under the accession number GSE106168.

RNA-seq affinity purification data have been deposited in the NCBI GEO database under accession number GSE106171.

LC-MS/MS data is available on the PRIDE Archive EMBL-EBI under the accession number PXD008128

## Acknowledgements

This project was supported by awards U19AI107792 (NIAID Functional Genomics Program) and 1R01GM087353 from the National Institutes of Health. We would like to thank members of the Crosson Lab for their contributions and input over the course of this project. The lab of Tao Pan provided important support in development of nucleic acid methods and lending equipment, most notably M.E. Evans and K.I. Zhou. Ruthenberg lab provided PP7 coat protein plasmids.

## Additional information

### Funding

| Funder | Grant reference number | Author |
|---|---|---|
| National Institutes of Health | 1R01GM087353 | Sean Crosson |
| National Institutes of Health | U19AI107792 | Sean Crosson |

The funders had no role in study design, data collection and interpretation, or the decision to submit the work for publication.

### Author contributions

Matthew Tien, Conceptualization, Software, Investigation, Methodology, Writing—original draft, Writing—review and editing; Aretha Fiebig, Conceptualization, Writing—original draft, Writing—review and editing; Sean Crosson, Conceptualization, Funding acquisition, Writing—original draft, Writing—review and editing

### Author ORCIDs

Matthew Tien http://orcid.org/0000-0002-0006-9644
Sean Crosson http://orcid.org/0000-0002-1727-322X

### Decision letter and Author response

Decision letter https://doi.org/10.7554/eLife.33684.043
Author response https://doi.org/10.7554/eLife.33684.044

## Additional files

### Supplementary files

• Supplementary file 1. Strains, plasmids, and primers used in this study.
DOI: https://doi.org/10.7554/eLife.33684.032
• Transparent reporting form
DOI: https://doi.org/10.7554/eLife.33684.033

### Major datasets

The following datasets were generated:

| Author(s) | Year | Dataset title | Dataset URL | Database, license, and accessibility information |
|---|---|---|---|---|
| Tien MZ, Fiebig A, Crosson S | 2017 | RNA-Seq gene expression analysis of ΔgsrN, gsrN-OE, and wild-type Caulobacter crescentus | https://www.ncbi.nlm.nih.gov/geo/query/acc.cgi?acc=GSE106168 | Publicly available at the NCBI Gene Expression Omnibus (accession no: GSE106168) |
| Tien MZ, Fiebig A, Crosson S | 2017 | Identification of RNAs that co-purify with the Caulobacter sRNA, GsrN | https://www.ncbi.nlm.nih.gov/geo/query/acc.cgi?acc=GSE106171 | Publicly available at the NCBI Gene Expression Omnibus (accession no: GSE106171) |
| Tien MZ, Fiebig A, Crosson S | 2017 | Caulobacter crescentus GsrN mutants proteome analysis | https://www.ebi.ac.uk/pride/archive/projects/PXD008128 | PXD008128 |

The following previously published dataset was used:

| Author(s) | Year | Dataset title | Dataset URL | Database, license, and accessibility information |
|---|---|---|---|---|
| Fang G, Passalacqua KD, Hocking J, Llopis PM, Gerstein M, Bergman NH, Jacobs-Wagner C | 2013 | Transcriptomic and phylogenetic analysis of a bacterial cell cycle reveals strong associations between gene co-expression and evolution | https://www.ncbi.nlm.nih.gov/geo/query/acc.cgi?acc=GSE46915 | Publicly available at the NCBI Gene Expression Omnibus (accession no: GSE46915) |

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
