## [Decision Letter]

Thank you for submitting your article "Gene network analysis identifies a central post-transcriptional regulator of cellular stress survival" for consideration by *eLife*. Your article has been reviewed by three peer reviewers, and the evaluation has been overseen by Michael Laub as the Reviewing Editor and Gisela Storz as the Senior Editor. The following individual involved in review of your submission has agreed to reveal his identity: Kai Papenfort.

The reviewers have discussed the reviews with one another and the Reviewing Editor has drafted this decision to help you prepare a revised submission.

In this manuscript, Tien et al., identified and characterized a new small regulatory (sRNA), called GsrN, in Caulobacter crescentus. Transcription of the gsrN gene is activated by the alternative σ factor, σ T, and GsrN is required for resistance against hydrogen peroxide and under hyperosmotic stress. In the cell, the GsrN sRNA accumulates in two isoforms: an unstable full-length variant and a stable 5' element, which is sufficient for hydrogen peroxide stress survival. Evidence is provided that generation of the stable 5' variant of the sRNA might require an endonucleolytic cleavage event (though see comments below). Using a 'pull-down' approach, the authors discover the katG mRNA as a direct interaction partner of GsrN and identify a base-pairing sequence involving five consecutive G-C base-pairs. GsrN increases KatG expression under regular as well as stress conditions, and this regulation might also be relevant in other stress settings. Overall, the article was well written, and the conclusions well supported by the data. The reviewers were particularly excited about the magnitude of the phenotype as few sRNAs in bacteria have strong null phenotypes. Although generally enthusiastic, the reviewers identified several issues that would need to be addressed in a revised manuscript

Essential revisions:

1) In other bacteria, many trans-acting sRNAs that regulate targets by short base pairing, these sRNA-mRNA interactions are mediated by Hfq or ProQ-like proteins. This is never really discussed or tested in the manuscript even though the GsrN-katG is suspiciously short, hinting at the potential requirement for such an RNA chaperone. The authors should interrogate available RNA-seq data of a Caulobacter Δhfq mutant from the Jacobs-Wagner lab (PMID:28827812) to at least comment on a potential Hfq dependence of the regulations reported here.

2) Table 1 and Figure 6: Given the G-C sequence stretch required for GsrN-katG pairing, I was wondering how many of the other predicted targets would have similar G-rich elements available for base-pairing? In general, the part on other potential targets of GsrN (subsection “GsrN is a general regulator of stress adaptation”) is a bit of a loose end, so the authors should substantially update/improve this section or consider deleting and developing it for presentation in a follow-up paper.

3) Subsection “GsrN is endonucleolytically processed into a more stable 5’ Isoform”, Figure 3 and Figure 3—figure supplement 1: I think these experiments are yet insufficient to determine the mechanism of how the two isoforms are generated. First, it is surprising that using a primer specific to the gsrN 3' end allows for the determination of the sRNA cleavage site in primer extension analysis, whereas no 3'end derived transcript was detected on Northern Blots. Second, the rifampicin experiments are not fully convincing, given that the 5S rRNA also decrease in the experiments. In general, it would be useful to confirm these data through additional experiments (e.g. differential 5' RACE experiments – e.g. see: PMID:8755555).

4) Figure 7 vs. B: The differences in katG mRNA vs. protein levels are difficult to understand: while the difference in katG mRNA levels are rather minor, KatG protein levels are >2-fold decreased in the gsrN mutant under steady state conditions. Conversely, treatment of WT cells increase katG mRNA levels by ~5-6 fold, while protein levels only change by ~1.5-fold. Please provide an explanation for this discrepancy.

5) There is no proposed model for how the interaction of GsrN basepairing to the 5' end of the katG mRNA (180 nt away from the translation start site?) can alter translation. This should be addressed.

---

## [Author Response]

Essential revisions:1) In other bacteria, many trans-acting sRNAs that regulate targets by short base pairing, these sRNA-mRNA interactions are mediated by Hfq or ProQ-like proteins. This is never really discussed or tested in the manuscript even though the GsrN-katG is suspiciously short, hinting at the potential requirement for such an RNA chaperone. The authors should interrogate available RNA-seq data of a Caulobacter Δhfq mutant from the Jacobs-Wagner lab (PMID:28827812) to at least comment on a potential Hfq dependence of the regulations reported here.

To address the role of Hfq, we added a supplementary figure to Figure 3 (Figure 3—supplementary figure 2). This figure reports results from experiments in which we probed GsrN isoform levels as a function of deletion and overexpression of *hfq*. Importantly *hfq* affects processing of GsrN: overexpression of *hfq* enhances the full-length isoform and deletion results in a depletion of full-length GsrN. We also added a short paragraph in our Results section to address this new piece of data.

We also analyzed the RNA-Seq data of PMID:28827812. However, the read-density at the *gsrN* locus in this study is low, and thus not useful for GsrN analysis. Low GsrN reads could be due to differences in RNA preparation techniques and/or media differences between our two studies. We used a defined minimal medium to conduct all of our phenotypic assays and extractions for northern blot analysis, while PMID:28827812 used complex peptone-yeast extract medium for their RNA-seq studies.

2) Table 1 and Figure 6: Given the G-C sequence stretch required for GsrN-katG pairing, I was wondering how many of the other predicted targets would have similar G-rich elements available for base-pairing? In general, the part on other potential targets of GsrN (subsection “GsrN is a general regulator of stress adaptation”) is a bit of a loose end, so the authors should substantially update/improve this section or consider deleting and developing it for presentation in a follow-up paper.

To address the number of other predicted targets that would have similar G-rich elements available for base-pairing we added a statement in our summary of the GsrN-PP7 RNA-Seq study. Furthermore, we have added a column to Table 1 that provides the predicted nucleotides that interact with GsrN from IntaRNA analysis. Already provided is the FASTA file (Figure 5—source data 3) of the predicted binding site of each entry in Table 1 so that readers may enter this file as well as GsrN’s sequence in the IntaRNA online tool.

3) Subsection “GsrN is endonucleolytically processed into a more stable 5’ Isoform”, Figure 3 and Figure 3—figure supplement 1: I think these experiments are yet insufficient to determine the mechanism of how the two isoforms are generated. First, it is surprising that using a primer specific to the gsrN 3' end allows for the determination of the sRNA cleavage site in primer extension analysis, whereas no 3'end derived transcript was detected on Northern Blots. Second, the rifampicin experiments are not fully convincing, given that the 5S rRNA also decrease in the experiments. In general, it would be useful to confirm these data through additional experiments (e.g. differential 5' RACE experiments – e.g. see: PMID:8755555).

To address the first statement “no 3’end derived transcript was detected on Northern Blots”, these smaller 3’ isoform products are seen in our blots with the 3’ probe, but at very low levels. The blot in Figure 3—figure supplement 1 is exposed to reveal these minor species. We have changed Figure 3—figure supplement 1 to more explicitly show the 3’isoform of GsrN. We also added a sentence to the main text to highlight this result. It is noteworthy that the size of these minor species is consistent with isoforms bearing the termini mapped by primer extension from the 3’ end and by 5’RACE (below).

Second, we agree that our rifampicin experiment is an indirect way to evaluate endonucleolytic processing of GsrN. Nevertheless, the results of this experiment, in particular the *increase* of the 5’ isoform after rifampicin treatment corresponds to a decrease of the full-length isoform, are consistent with a model of processing.

Finally, per the reviewer’s suggestion, we conducted differential 5’RLM-RACE. We were able to obtain 5’RACE products with termini that correspond to the two 3’ isoforms observed in Figure 3—figure supplement 1 in untreated (no TAP) total RNA samples from *gsrN^++^*. Their presence in no-TAP samples supports a model of an endonucleolytic cleavage origin for the termini. We summarized our results in Figure 3—figure supplement 1. Together, multiple lines of evidence support the existence of 3’ isoforms generated by endonucleolytic processing.

4) Figure 7 vs. B: The differences in katG mRNA vs. protein levels are difficult to understand: while the difference in katG mRNA levels are rather minor, KatG protein levels are >2-fold decreased in the gsrN mutant under steady state conditions. Conversely, treatment of WT cells increase katG mRNA levels by ~5-6 fold, while protein levels only change by ~1.5-fold. Please provide an explanation for this discrepancy.

The reviewer raises an important point. We have added an additional paragraph to discuss these trends. *katG* mRNA is induced ~5-fold by hydrogen peroxide in both wild-type and *gsrN* mutants; we attribute this to OxyR-dependent activation (independent of *gsrN*). Although, *katG* transcript and protein levels do not scale 1:1, both are induced by hydrogen peroxide in wild-type and *gsrN^++^* strains. In cells lacking *gsrN, katG* transcript is induced by peroxide yet KatG protein does not change significantly. This suggests that translation of *katG* mRNA is sensitive to GsrN.

Given the 15-minute treatment, we are not sure that one should expect the same fold change in message and protein levels. Our results may reflect that transcript accumulates more quickly than protein, and/or that the *katG* message is not translated efficiently.

5) There is no proposed model for how the interaction of GsrN basepairing to the 5' end of the katG mRNA (180 nt away from the translation start site?) can alter translation. This should be addressed.

To address this point, we emphasize how other sRNAs in that bind the 5’UTR of their mRNA targets activate expression. Specifically, we cite how RydC and FasX stabilize their mRNA targets, RepG blocks stand-by ribosome sequences upstream of the SD, and VR-RNA stabilizes *colA* through processing the 5’UTR which stabilizes the transcript.